# 5-FU promotes stemness of colorectal cancer via p53-mediated WNT/β-catenin pathway activation

Yong-Hee Cho [1,6], Eun Ji Ro [1,6], Jeong-Su Yoon[1], Tomohiro Mizutani [2], Dong-Woo Kang [3], Jong-Chan Park[1], Tae Il Kim [4], Hans Clevers [2] & Kang-Yell Choi [1,5 ✉]

5-Fluorouracil (5-FU) remains the first-line treatment for colorectal cancer (CRC). Although 5-FU initially de-bulks the tumor mass, recurrence after chemotherapy is the barrier to effective clinical outcomes for CRC patients. Here, we demonstrate that p53 promotes *WNT3* transcription, leading to activation of the WNT/β-catenin pathway in *Apc^{Min/+}/Lgr5^{EGFP}* mice, CRC patient-derived tumor organoids (PDTOs) and patient-derived tumor cells (PDCs). Through this regulation, 5-FU induces activation and enrichment of cancer stem cells (CSCs) in the residual tumors, contributing to recurrence after treatment. Combinatorial treatment of a WNT inhibitor and 5-FU effectively suppresses the CSCs and reduces tumor regrowth after discontinuation of treatment. These findings indicate p53 as a critical mediator of 5-FU-induced CSC activation via the WNT/β-catenin signaling pathway and highlight the significance of combinatorial treatment of WNT inhibitor and 5-FU as a compelling therapeutic strategy to improve the poor outcomes of current 5-FU-based therapies for CRC patients.

[1] Department of Biotechnology, College of Life Science and Biotechnology, Yonsei University, Seoul 03722, Korea. [2] Hubrecht Institute, Cancer Genomics Netherlands, UMC Utrecht, 3584CT Utrecht, Netherlands. [3] Medpacto Bio Institute, Medpacto Inc, Seoul 06668, Republic of Korea. [4] Division of Gastroenterology, Department of Internal Medicine, Yonsei University College of Medicine, Seoul 03722, Korea. [5] CK Biotechnology Inc, Yonsei Engineering Complex B137A, 50 Yonsei Ro, Seodaemun-Gu, Seoul 03722, Korea. [6] These authors contributed equally: Yong-Hee Cho, Eun Ji Ro. ✉email: kychoi@yonsei.ac. kr

Colorectal cancer (CRC) is one of the most common cancers worldwide[1]. The fluorinated analog of uracil, 5-fluorouracil (5-FU), is a fundamental component of chemotherapeutic agents for palliative and adjuvant treatments of CRC[2,3]. The clinical benefits of 5-FU treatment, however, are often short-lived, and the majority of treated patients do not realize complete eradication of tumor cells, resulting in poor outcomes due to recurrence after 5-FU therapy[4]. 5-FU-based therapies, such as FOLFOX (5-FU, leucovorin, and oxaliplatin) or FOLFIRI (5-FU, leucovorin, and irinotecan), have been used as the standard therapy for advanced CRC[5]. Although 5-FU-based therapies have increased the objective response rates to 40–50% in CRC patients, the disease-free survival of CRC patients has not effectively been extended[6,7]. Therefore, strategies for improving the clinical outcomes of 5-FU treatments are urgently needed, and understanding the mechanisms by which relapse occurs in 5-FU-treated CRC patients is an essential step toward increasing the survival benefit of 5-FU-based treatment for CRC.

Clinical benefits of chemotherapies are often associated with cancer stem cells (CSCs)[8]. CSCs are a small population of cancer cells that undergo continuous self-renewal and differentiate into heterogeneous cells, yielding tumor-initiating potential. For this reason, the presence of CSCs permits the relapse of tumors treated with chemotherapy[9]. Clinical studies of advanced CRC patients have reported that high expression levels of CSC markers correlate with poor overall survival rates after chemotherapies[10,11]. In addition, accumulating evidence suggests that eradication of CSCs, which are enriched by 5-FU-based therapies, is a potential approach to overcome chemotherapy failure via prevention of tumor recurrence[12–14]. However, the key regulatory mechanisms responsible for the increased stemness under chemotherapy remain largely unknown.

Wnt/β-catenin signaling pathway is a critical regulator of the homeostasis of normal intestinal stem cells (ISCs)[15]. Secretion of WNT3 from ISC neighboring cells and ISC-specific expression of R-spondin1 establish the WNT/β-catenin gradient along the crypt-villus axis, controlling upward movement of progeny cells from the crypt bottom toward the villus and differentiation into all cell types that constitute the intestine[16–18]. In CRC, aberrant activation of the WNT/β-catenin signaling pathway, mostly attributed to *adenomatous polyposis coli* (*APC*) mutations that occur in 90% of CRC patients, disrupts the WNT gradient, allows hyperplasia of ISCs along the crypt-villus axis, and drives tumor initiation via CSCs[19]. The WNT/β-catenin signaling pathway also plays essential roles in the activation and maintenance of colorectal CSCs during progression of CRCs, and activation of this pathway is a hallmark of poor CRC patient prognosis[20,21]. Despite its critical roles in CSC enrichment in CRC, the involvement of WNT/β-catenin signaling in the recurrence after chemotherapy treatment has not yet been investigated.

The sequence-specific transcription factor p53 is a key player in the cellular response to stress via its role in inhibiting proliferation or inducing programmed cell death. Furthermore, p53 serves as the major route for anti-cancer effect of 5-FU and determines the cellular sensitivity to cytotoxic 5-FU[22–24]. 5-FU treatment increases translation, stability and transcriptional activity of p53 and target genes of p53 promote various anti-cancer processes such as cell cycle arrest, apoptosis and senescence[25]. Despite intense efforts on identification of the association of p53 and 5-FU response[5,26,27], the involvement of the tumor suppressor p53 in the relapse after 5-FU treatment has not been investigated yet. Emerging evidence suggests that p53 cooperates with other signaling pathways to play additional roles in various biological processes[28,29]. Based on these recent findings showing p53 has dual functions in regulating both death and survival of mouse embryonic stem cells (ESCs) via transcriptional activation of Wnt pathway[28,29], we have investigated an additional role of p53 in CSC activation in 5-FU-treated CRC.

In this study, we demonstrate that 5-FU activates CSCs via p53-induced *WNT3* transcription, which is followed by activation of the WNT/β-catenin pathway in CRC cell lines and xenograft tumors as well as patient avatar models, such as patient-derived tumor organoids (PDTOs) and patient-derived cells (PDCs). Critical roles of the WNT/β-catenin pathway on the activation and enrichment of CSCs by 5-FU are demonstrated by the CSC-suppressive effects of the combinatorial treatment of a WNT inhibitor with 5-FU. In addition, combinatorial treatments of the WNT inhibitor and 5-FU inhibit the regrowth of PDTOs and xenograft tumors after discontinuation of the treatments. Taken together, our data uncover a direct and functional connection between p53 and the WNT/β-catenin signaling pathway and highlight their involvement in CSC activation by 5-FU treatment in CRC. Furthermore, these results suggest that combinatorial treatment of CRC patients with a WNT inhibitor and 5-FU can be an effective strategy to overcome poor survival rates following 5-FU-based therapy for CRC.

## Results

**5-FU induces enrichment of Lgr5+ cells.** Despite the increased sensitivity of 5-FU-based therapy, such as FOLFOX or FOLFIRI, compared to treatment with only 5-FU, patient survival after treatment still remains poor in total (Supplementary Fig. 1a) and different stages of CRCs (Supplementary Fig. 1b–d). To understand the fundamental issues of 5-FU-based therapy associated with poor prognosis, we investigated whether 5-FU treatment enriches CSCs in CRC. Treatment with 5-FU effectively reduced both numbers and sizes of tumor organoids derived from intestinal tumors of $Apc^{Min/+}/Lgr5^{EGFP}$ mice and significantly suppressed their growth (Fig. 1a, b); however, in the residual tumor organoids, the intensity of the CSC marker Lgr5 was significantly increased as monitored by GFP signal (Fig. 1c). Activation of CSCs by 5-FU was confirmed by significant increases in the mRNA expression levels of *Lgr5*, *Cd44*, *Cd133*, and *Cd166* (Fig. 1d). Time-course analysis of Lgr5 expression in tumor organoids by analysis of GFP intensity showed that as the tumor organoids grew, CSC activation did not change. However, 5-FU treatment increased activation of CSCs while suppressing the growth of tumor organoids, as shown by the increase in the GFP intensity at 48 h treatment (Fig. 1e; the gradual change is shown in Supplementary Movies 1 and 2). Enrichment and activation of CSCs by 5-FU were also confirmed in vivo by immunohistochemical (IHC) analysis of Lgr5 (GFP) in 5-FU treated $Apc^{Min/+}/Lgr5^{EGFP}$ mouse tumors (Fig. 1f).

**5-FU activates the Wnt/β-catenin pathway in CRC.** To identify the mechanism by which 5-FU leads to CSC activation and enrichment, we assessed the status of the Wnt/β-catenin signaling pathway. Treatment with 5-FU significantly increased both total and active β-catenin levels in tumor organoids derived from $Apc^{Min/+}/Lgr5^{EGFP}$ mice, as shown by immunoblotting and immunocytochemical (ICC) analyses (Fig. 2a, b). The activation of the Wnt/β-catenin signaling pathway by 5-FU was also confirmed by a significant increase in the β-galactosidase staining of 5-FU-treated tumor organoids derived from the Wnt reporter mouse $Axin2^{LacZ}$[30], crossed with $Apc^{Min/+}$ mice ($Apc^{Min/+}/Axin2^{LacZ}$; Fig. 2c). Time-course analysis of β-catenin expression after treatment with 5-FU revealed a significant increase in β-catenin levels after 12 h of treatment, suggesting that activation of the Wnt/β-catenin signaling pathway occurs prior to the activation and enrichment of CSCs by 5-FU (Figs. 1e and 2d). In addition, IHC analyses showed that β-catenin levels were

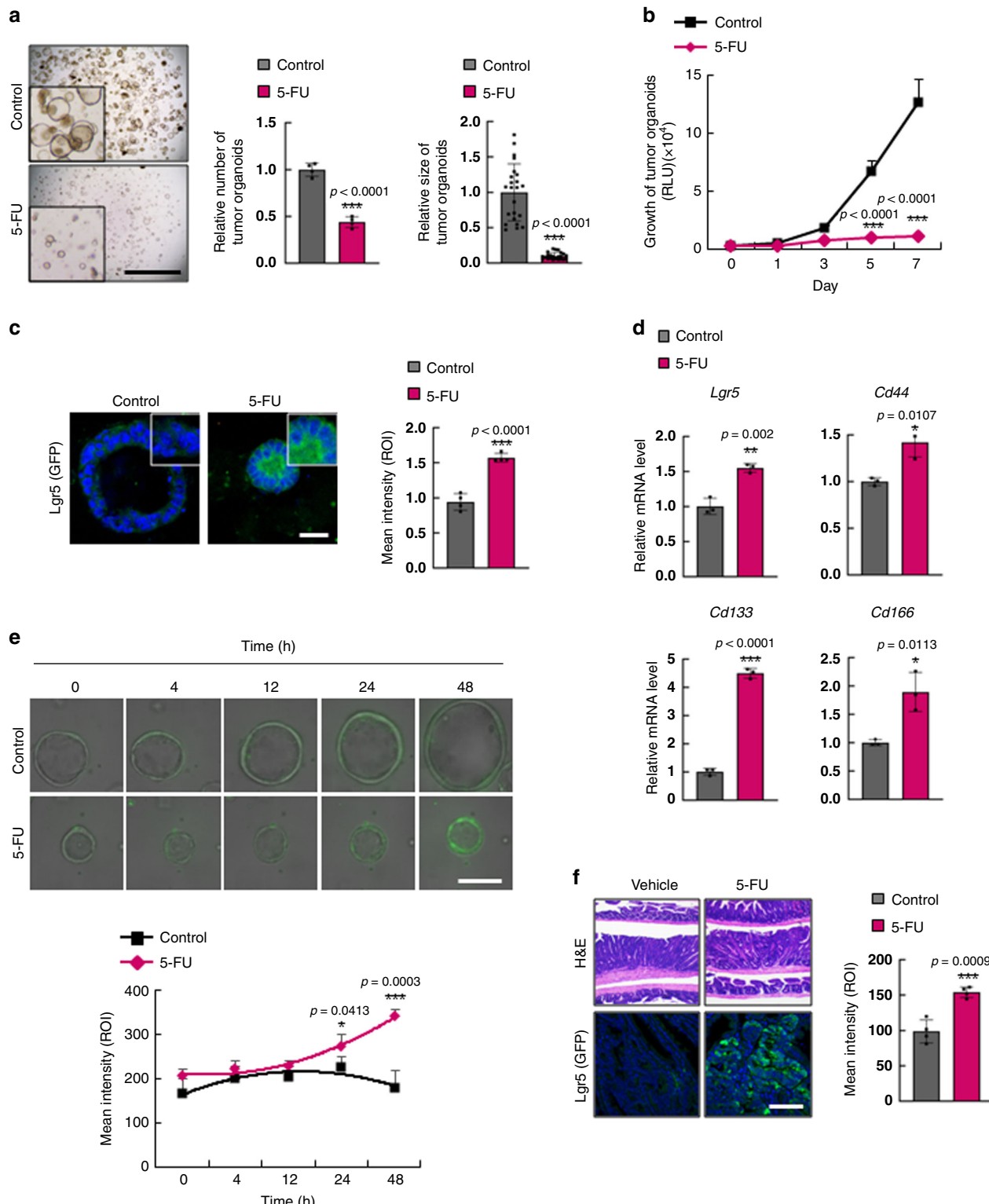

significantly elevated in the tumors of 5-FU-treated $Apc^{Min/+}/$ $Lgr5^{EGFP}$ mice, and that these levels were strongly correlated with the expression of Lgr5 (Fig. 2e, f). Moreover, 5-FU treatment of mouse tumor organoids derived from $Apc^{Min/+}/K\text{-}Ras^{G12D}LA2/$ $Lgr5^{EGFP}$ mice and in vivo injection of 5-FU to $Apc^{Min/+}/K\text{-}$ $Ras^{G12D}LA2/Lgr5^{EGFP}$ mice confirmed increased expression of β-catenin and Lgr5 (GFP), showing that 5-FU-induced activation of the Wnt/β-catenin signaling pathway and CSCs also occur in an advanced murine CRC model (Supplementary Fig. 2a, b).

**5-FU activates the WNT/β-catenin pathway via p53 in CRC.** We next sought to explore the mechanism by which 5-FU activates the WNT/β-catenin signaling pathway in CRC. We first confirmed the activation of the WNT/β-catenin signaling pathway by 5-FU treatment in various human CRC cell lines. Interestingly, 5-FU increased β-catenin levels in CRC cell lines harboring wild-type *p53* (p53 WT) but not in those harboring mutant *p53* (p53 MUT; Fig. 3a). The ICC analyses revealed a positive correlation between the expressions of p53 and β-catenin

**Fig. 1 5-FU treatment induces CSC enrichment in murine CRC. a–f** Analyses of tumor organoids derived from intestinal tumors of $Apc^{Min/+}/Lgr5^{EGFP}$ mice after treatment with 5-FU (1.5 μg/ml, 48 h). **a** Bright-field images of tumor organoids treated with 5-FU. Relative numbers ($n = 4$ biologically independent samples per group) and sizes ($n = 23$ biologically independent organoids per group) of tumor organoids were quantified as fold-change compared to control. Scale bar = 5 mm. **b** Growth of tumor organoids after treatment with 5-FU for indicated days. ($n = 5$ biologically independent samples per group) **c** Confocal images of Lgr5 (GFP) and the mean intensity of GFP after treatment with 5-FU. Scale bar = 50 μm. ($n = 5$ biologically independent samples per group.) **d** Relative mRNA levels of $Lgr5$, $Cd44$, $Cd133$, and $Cd166$ after 5-FU treatment. Data shown as fold-change compared to control. ($n = 3$ biologically independent samples per group.) **e** Time-course analyses of Lgr5 (GFP) expression after 5-FU treatment. Scale bar = 100 μm. ($n = 4$ biologically independent samples per group.) **f** Hematoxylin and eosin and immunofluorescence staining of mouse intestinal sections after treatment with vehicle or 5-FU (25 kg/ml, 3 weeks) using antibodies for the indicated proteins. The mean intensities were measured by Zen software 3.1. Scale bar = 50 μm. ($n = 4$ biologically independent samples per group) Data are mean ± s.d., two-sided Student's $t$-test, *$p < 0.05$, **$p < 0.01$, ***$p < 0.001$. Source data are provided as a Source data file.

in 5-FU-treated CRC cell lines (Fig. 3b). To assess the involvement of p53 in the 5-FU-induced activation of the WNT/β-catenin signaling pathway, we determined the effect of 5-FU on the WNT/β-catenin signaling pathway in HCT116 p53 wild-type ($p53^{+/+}$), null ($p53^{-/-}$), and mutant ($p53^{R248W/-}$) isogenic human CRC cell lines. 5-FU treatment increased β-catenin levels in $p53^{+/+}$ cells but not in $p53^{-/-}$ and $p53^{R248W/-}$ cells (Fig. 3c). We then investigated the involvement of p21$^{Waf1/Cip1}$ (p21), the downstream p53 target gene that mediates p53-induced cell cycle arrest, in the 5-FU-induced activation of the WNT/β-catenin signaling pathway using HCT116 p21 wild-type ($p21^{+/+}$) and null ($p21^{-/-}$) isogenic human CRC cell lines. The 5-FU-induced increase in β-catenin activation occurred regardless of the $p21$ status (Fig. 3d), indicating that p53 mediates 5-FU-induced activation of the WNT/β-catenin signaling pathway independently of its role in cell cycle arrest. Pifithrin-α, a p53 inhibitor that blocks the transcriptional activity of p53 suppressed the 5-FU-induced β-catenin increase, and contrarily, the p53 activator RITA increased β-catenin level in the HCT116 human CRC cell line (Fig. 3e, f). The increase in β-catenin by p53 was also confirmed in tumor organoids derived from $Apc^{Min/+}/Lgr5^{EGFP}$ mice (Fig. 3g, h), clearly showing that p53 activity regulates the WNT/β-catenin signaling pathway in CRC.

**5-FU induces WNT3 transcription via p53.** Given that CSCs share characteristics with ISCs, we investigated whether the most essential niche factors WNT3 and R-spondin1 for maintaining the activation of the WNT/β-catenin pathway of ISCs were involved in the p53-mediated activation of the WNT/β-catenin pathway and CSC activation in 5-FU-treated CRC. Virtual prediction identified p53 as one of the top putative transcription factors for WNT3, and $WNT3$ was identified as a direct target of p53 in the presence of DNA damages in embryonic stem cell[28,29].

Therefore, we assessed whether p53 can regulate transcription of WNT3 in CRC cells. Indeed, the treatment of 5-FU increased the mRNA level of Wnt3 in tumor organoids derived from $Apc^{Min/+}/Lgr5^{EGFP}$ mice (Fig. 4a). The role of p53 in 5-FU-induced $WNT3$ transcription in human CRC was also shown by increased $WNT3$ mRNA levels in human CRC cell lines harboring wild-type $p53$ but not in those harboring mutant $p53$ (Fig. 4b). Treatment of HCT116 isogenic cell lines with either wild-type ($p53^{+/+}$), null ($p53^{-/-}$), and mutant ($p53^{R248W/-}$) $p53$ and wild-type ($p21^{+/+}$) and null ($p21^{-/-}$) $p21$ with 5-FU showed that the 5-FU-induced $WNT3$ transcription required $p53$ but occurred independently of the status of $p21$ (Fig. 4c, d). Consistently, co-treatment with pifithrin-α and 5-FU inhibited the 5-FU-induced increase in $WNT3$ mRNA levels (Fig. 4e). In contrast, treatment of RITA induced $WNT3$ transcription in the HCT116 human CRC cell line (Fig. 4f). Since the 5-FU-induced $WNT3$ transcription was abrogated by the DNA-binding mutation of p53, we speculated that p53 may function as a direct transcription factor for $WNT3$ induction. Indeed, the p53

interaction with the $WNT3$ promoter was enhanced upon 5-FU treatment in HCT116 human CRC cells as shown by chromatin immunoprecipitation (ChIP) analysis (Fig. 4g). Increased expression of both WNT3 and β-catenin by 5-FU were observed via ICC analyses in human intestinal organoids with $APC$ knockout and wild-type $p53$ ($APC^{KO}/p53^{WT}$) but not in those with $APC$ knockout and mutant $p53$ ($APC^{KO}/p53^{KO}$)[31] (Fig. 4h). Together, these results indicate that 5-FU induces $WNT3$ via p53-mediated transcriptional activation.

**WNT inhibition suppresses 5-FU-induced CSC activation.** Consistent with the p53-dependent intracellular WNT3 increase, secretion of WNT3 was significantly increased by treatment of 5-FU in $p53^{+/+}$ HCT116 cells but not in the $p53^{-/-}$ HCT116 cells (Supplementary Fig. 3a). Treatment of exogenous WNT3 increased both total and active β-catenin levels, together with increases in the mRNA levels of CSC markers in $p53$ wild-type CRC cell lines, HCT116 and LoVo cells (Supplementary Fig. 3b, c).

To explore whether WNT3 induction is responsible for the activation of the WNT/β-catenin signaling pathway and subsequent CSC activation in 5-FU-treated CRC, we treated tumor organoids derived from $Apc^{Min/+}/Lgr5^{EGFP}$ mice with porcupine inhibitor LGK-974[32] and 5-FU. The increases in the expression levels of β-catenin and Lgr5 (GFP) induced by 5-FU were effectively inhibited by LGK-974 as shown by ICC analyses (Fig. 5a). After 48 h of 5-FU alone or combinatorial treatment with LGK-974, the time point when cancer stem cell markers are effectively induced by 5-FU and those inductions were inhibited by combinatorial treatment with LGK-974 (Figs. 1e and 5a), tumor organoids were dissociated and reseeded with fresh media without drug treatments. Here, initiation of organoid clone formation derived from 5-FU-treated $Apc^{Min/+}/Lgr5^{EGFP}$ tumor organoids occurred earlier with faster growth rate than that derived from 5-FU and LGK-974 co-treated $Apc^{Min/+}/Lgr5^{EGFP}$ tumor organoids. (Supplementary Fig. 4a–c), showing that WNT-induced activation of CSCs by 5-FU is associated with functional characteristics of colorectal CSCs. Effective suppression of 5-FU-induced WNT3 secretion (Fig. 5b) and inhibition of 5-FU-induced activation of the WNT/β-catenin signaling pathway and CSC were observed following LGK-974 co-treatment with 5-FU in the HCT116 human CRC cell line, as demonstrated by protein and mRNA levels of β-catenin and CSC markers, respectively (Fig. 5c, d). The mechanism of p53-mediated transcriptional induction of $WNT3$ followed by activation of the WNT/β-catenin signaling pathway and subsequent activation of CSCs was also confirmed in the LoVo CRC cell line, which harbors wild-type $p53$ (Supplementary Fig. 5a–i).

We then investigated the in vivo effects of combined treatment with LGK-974 and 5-FU using an HCT116 cell line xenograft mouse model. To investigate the effects of WNT inhibition on recurrence after 5-FU, we intentionally stopped the treatment and

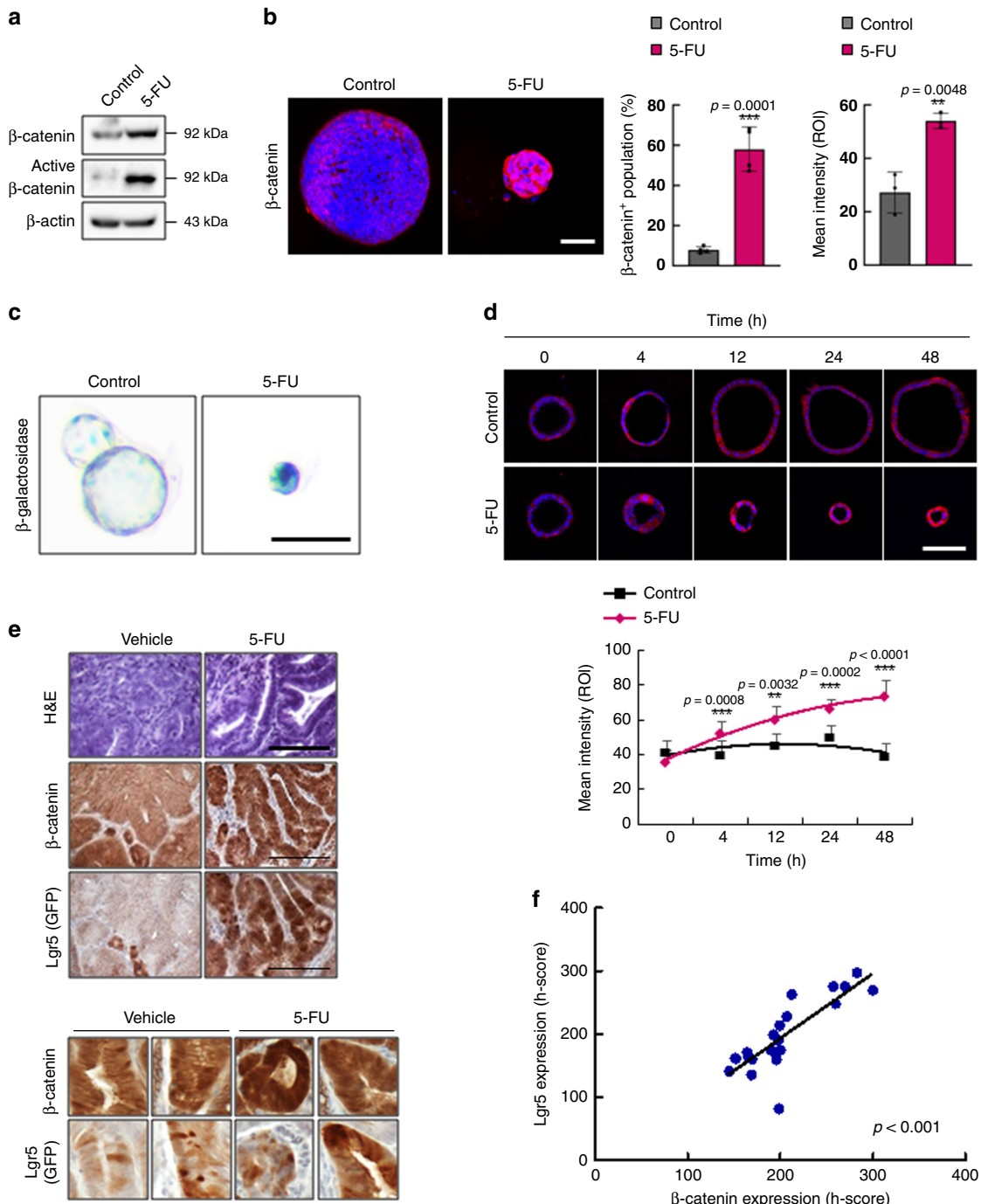

**Fig. 2 5-FU activates the WNT/β-catenin pathway in murine CRC. a** Immunoblots of indicated proteins in $Apc^{Min/+}/Lgr5^{EGFP}$ tumor organoids with or without 5-FU treatment (1.5 μg/ml, 48 h). **b** Confocal images of β-catenin (red) in $Apc^{Min/+}/Lgr5^{EGFP}$ tumor organoids with or without 5-FU treatment. The percentage of β-catenin$^+$ cells ($n = 4$ biologically independent samples per group) and the mean intensities ($n = 3$ biologically independent samples per group) were quantified. Scale bar = 50 μm. **c** β-galactosidase staining of the tumor organoids derived from $Apc^{Min/+}/Axin2^{LACZ}$ mice with or without 5-FU treatment. Scale bar = 100 μm. **d** Time-course analysis of β-catenin expression in $Apc^{Min/+}/Lgr5^{EGFP}$ tumor organoids with or without 5-FU treatment and quantification of mean intensities. Scale bar = 100 μm. **e, f** $Apc^{Min/+}/Lgr5^{EGFP}$ mice were treated with vehicle or 5-FU (25 mg/kg, 3 weeks). **e** Hematoxylin and eosin staining and DAB staining of β-catenin and Lgr5 (GFP) in sections of intestinal tumors of $Apc^{Min/+}/Lgr5^{EGFP}$ mice treated with vehicle or 5-FU. Scale bar = 100 μm. **f** Linear regression curve showing Lgr5 and β-catenin expression in tumors of $Apc^{Min/+}/Lgr5^{EGFP}$ mice treated with 5-FU. The mean intensities were measured by Zen software 3.1. ($n = 21$ fields examined over four biologically independent animals per group) Data are mean ± s.d., two-sided Student's $t$-test, **$p < 0.01$, ***$p < 0.001$. Source data are provided as a Source data file.

allowed the tumors to regrow in the absence of treatment. Both treatment with 5-FU alone and co-treatment with the WNT inhibitor LGK-974 and 5-FU effectively reduced the tumor growth initially. Intriguingly, after discontinuation of treatment,

5-FU-treated tumors rapidly regrow and reached tumor sizes similar to those in the untreated group, while tumors co-treated with LGK-974 and 5-FU exhibited suppressed tumor regrowth after discontinuation of treatment (Fig. 5e, f). At the time of

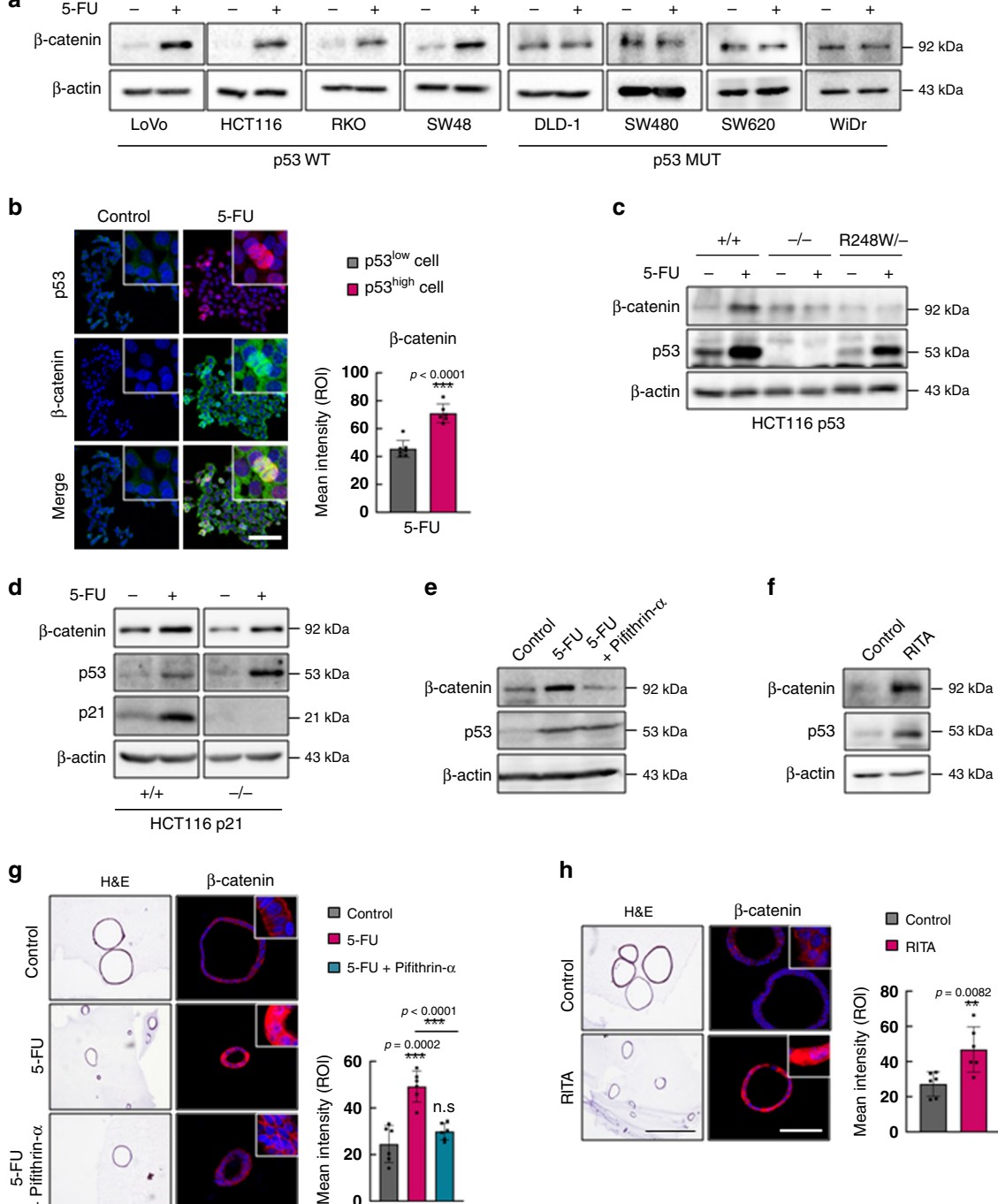

**Fig. 3 p53 mediates 5-FU-induced activation of the WNT/β-catenin signaling pathway. a** Immunoblots of indicated proteins in various CRC cell lines treated with or without 5-FU (1.5 μg/ml, 48 h). **b** Confocal images of immunofluorescence staining of HCT116 cells treated with or without 5-FU and the mean intensity of β-catenin in p53low and p53high cells. Scale bar = 20 μm. (p53low $n = 7$, p53high $n = 6$ biologically independent samples). **c** Immunoblots of indicated proteins in $p53^{+/+}$, $p53^{-/-}$, and $p53^{R248W/-}$ isogenic HCT116 cell lines treated with or without 5-FU. **d** Immunoblots of indicated proteins in $p21^{+/+}$ and $p21^{-/-}$ isogenic HCT116 cell lines treated with or without 5-FU. **e** Immunoblots of indicated proteins in HCT116 cells treated with or without treatment of 5-FU and pifithrin-α (10 μM, 48 h). **f** Immunoblots of indicated proteins in HCT116 cells treated with or without RITA (5 μM, 48 h). **g, h** Hematoxylin and eosin staining and immunofluorescence staining of β-catenin in $Apc^{Min/+}/Lgr5^{EGFP}$ intestinal tumor organoids treated with or without 5-FU and pifithrin-α (**g**) and with or without RITA (**h**). Scale bar = 50 μm. The mean intensities were measured by Zen software 3.1. ($n = 6$ biologically independent samples per group) Data are mean ± s.d., two-sided Student's $t$-test, n.s. not significant, *$p < 0.05$, **$p < 0.01$, ***$p < 0.001$. Source data are provided as a Source data file.

treatment discontinuation, despite the reduction in tumor size, tumors treated with 5-FU had higher protein levels of β-catenin and higher mRNA levels of the CSC markers *LGR5*, *CD44*, *CD133*, and *CD166* than the untreated tumors; however, in the tumors co-treated with LGK-974 and 5-FU, the levels of β-catenin and CSC markers were not increased compared to untreated controls. (Fig. 5g, h). IHC analyses of the tumors confirmed the elevated expression of β-catenin as well as the

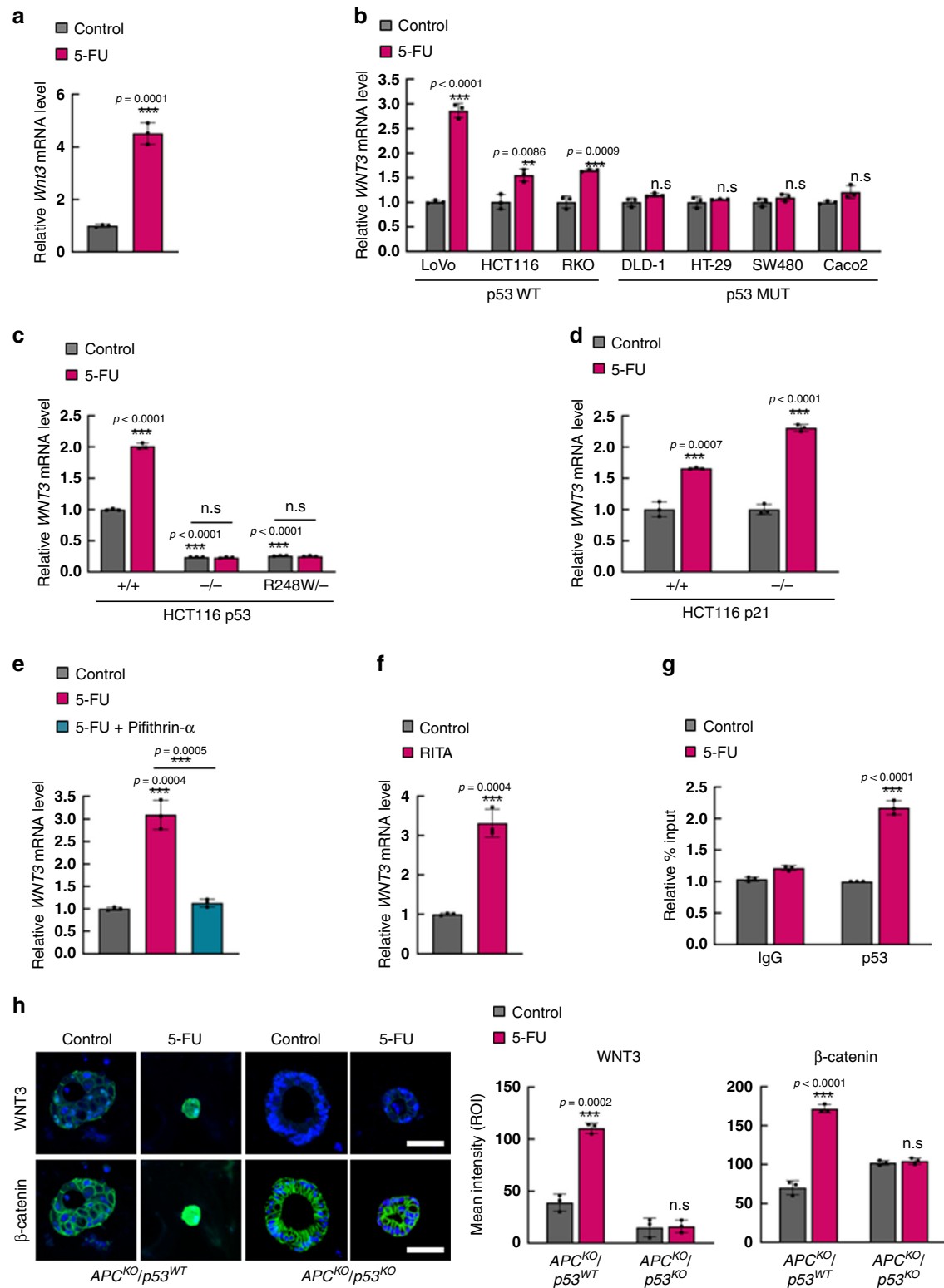

CD44, CD133, and CD166 CSC markers by 5-FU treatment and the inhibitory effects of LGK-974 on the 5-FU-induced activation of the WNT/β-catenin signaling and subsequent CSC activation (Fig. 5i). Therefore, CSC activation via p53-mediated *WNT3* induction is important for the 5-FU-induced recurrence of CRC.

**WNT inhibition suppresses the regrowth after 5-FU treatment.** To investigate the clinical relevance of WNT3-mediated CSC activation on recurrence in CRC patient after 5-FU treatment, we

next examined the effects of combinatorial treatment with LGK-974 and 5-FU on CRC PDCs[33] and PDTOs. Treatment with 5-FU significantly increased the levels of p53, WNT3, and β-catenin in PDCs and PDTOs that harbored wild-type *p53* (Supplementary Table 1 and Fig. 6a), and the mRNA levels of *WNT3* were also increased by 5-FU treatment (Fig. 6b). Consistently, the mRNA levels of *LGR5* were also increased by treatment with 5-FU in both PDCs and PDTOs with wild-type *p53* (Fig. 6c); however, 5-FU-induced activation of the WNT/β-catenin signaling pathway

**Fig. 4 5-FU activates WNT/β-catenin signaling via p53-dependent *WNT3* transcrption. a** Relative mRNA levels of *Wnt3* in *Apc$^{Min/+}$/Lgr5$^{EGFP}$* mouse tumor organoids after treatment with 5-FU (1.5 µg/ml, 48 h). (*n* = 3 biologically independent samples per group). **b** Relative mRNA levels of *WNT3* in various CRC cell lines after 5-FU treatment. (*n* = 3 biologically independent samples per group). **c** Relative mRNA levels of *WNT3* in *p53$^{+/+}$*, *p53$^{-/-}$*, and *p53$^{R248W/-}$* isogenic HCT116 cell lines after 5-FU treatment. (*n* = 3 biologically independent samples per group). **d** Relative mRNA levels of *WNT3* in *p21$^{+/+}$* and *p21$^{-/-}$* isogenic HCT116 cell lines after 5-FU treatment. (*n* = 3 biologically independent samples per group). **e** Relative mRNA levels of *WNT3* in HCT116 cells treated with or without 5-FU and pifithrin-α (10 µM, 48 h). (*n* = 3 biologically independent samples per group). **f** Relative mRNA level of *WNT3* in HCT116 cells treated with our with RITA (5 µM, 48 h). (*n* = 3 biologically independent samples per group). **g** Relative percentage input of ChIP-qPCR analysis of *WNT3* promoter using indicated antibodies in HCT116 cells after treatment with 5-FU. (*n* = 3 biologically independent samples per group) **h** Confocal images of WNT3 (green) and β-catenin (green) with DAPI in *APC$^{KO}$/p53$^{WT}$* and *APC$^{KO}$/p53$^{KO}$* human colon organoids treated with or without 5-FU. Scale bar = 50 µm. Quantification of mean intensities of WNT3 and β-catenin in Fig. 4h. The mean intensities were measured by Zen software 3.1. (*n* = 3 biologically independent samples per group) Data are mean ± s.d., two-sided Student's *t*-test, n.s. not significant, *$p < 0.05$, **$p < 0.01$, ***$p < 0.001$. Source data are provided as a Source data file.

followed by CSC enrichment did not occur in the PDC harboring mutant *p53* (Supplementary Fig. 6a, b) confirming the p53-dependent activation of CSCs by 5-FU treatment in CRC patient. Furthermore, 5-FU-induced increase in β-catenin was effectively suppressed by co-treatment with LGK-974 in PDCs and PDTOs (Fig. 6d, e), confirming the involvement of WNT ligand in the 5-FU-mediated activation of the WNT/β-catenin signaling in CRC patients. To further determine the pathophysiological importance of WNT inhibition in vivo, we assessed the effects of combinatorial treatment of 5-FU and LGK-974 in a *p53* wild-type PDC xenograft mouse model. Both 5-FU treatment alone and co-treatment with LGK-974 with 5-FU effectively reduced tumor growth, and co-treatment increased the sensitivity of the tumors to 5-FU (Fig. 6f and Supplementary Fig. 6c). Treatment with 5-FU alone, however, increased β-catenin and CSC markers in the remaining tumor, while concurrent LGK-974 treatment effectively suppressed the effects of 5-FU on the levels of β-catenin and CSC markers (Supplementary Fig. 6d, e and Fig. 6g). Together, these data validate the key role of WNT induction in CSC activation by 5-FU in vivo.

We then explored the efficacy of combined treatment with LGK-974 and 5-FU on tumor recurrence after 5-FU treatment using CRC PDTO harboring wild-type *p53*. Consistently, both 5-FU treatment alone and LGK-974 co-treatment effectively reduced the growth of PDTO. After discontinuation of treatment; however, 5-FU-treated PDTO rapidly regrew and reached sizes similar to those in untreated PDTOs, while PDTOs that were co-treated with LGK-974 and 5-FU showed markedly reduced regrowth after discontinuation of treatment (Fig. 6h, i). Collectively, our in vitro, in vivo, and ex vivo results demonstrated a critical role for *WNT3* induction in mediating CSC activation and recurrence in 5-FU-treated CRCs with wild-type *p53*. Despite the initial tumor de-bulking, the 5-FU-treated tumors regrew due to CSC activation via p53-mediated transcriptional *WNT3* induction. Similar to 5-FU, oxaliplatin (OXA) and irinotecan (IRI), the alternative p53 activating chemotherapies for CRC patients[34–37], significantly increased the β-catenin level and expressions of WNT3 and CSC markers (Supplementary Fig. 7a–c). These drug effects were mostly abolished by LGK-974 co-treatment in CRC cell lines harboring wild-type *p53* (Supplementary Fig. 7d–g). In addition, 5-FU-resistant (5-FU-R) cell line with wild-type *p53*, but not those with mutant *p53* exhibited elevated β-catenin and WNT3 levels compared to their parental counterparts (Supplementary Fig. 8a, b). Consistent with the WNT/β-catenin pathway activation, the increased spheroid-forming ability and induction of CSC markers in the 5-FU-R-HCT116 cells were inhibited by LGK-974, but not by DAPT, the NOTCH inhibitor (Supplementary Fig. 8c–e). However, the cell viability inhibition of 5-FU-R-HCT116 cells by co-treatment of LGK-974 with 5-FU (23.4%) showed partial effect (Supplementary Fig. 8f, g), compared with that of parental

HCT116 cells by treatment of 5-FU (73.5%) (Supplementary Fig. 8a). These results indicate that the mechanism that we identified by transient treatment of 5-FU plays crucial roles in the activation of CSCs in both 5-FU-sensitive and -resistant *p53* wild-type CRC cells, but other driving factors might be involved in the 5-FU-resistance. Together, combinatorial treatment with 5-FU and a WNT inhibitor, which blocks WNT ligand-induced activation of the WNT/β-catenin pathway and suppresses 5-FU-induced activation and enrichment of CSCs, effectively reduced tumor regrowth following initial 5-FU treatment, overcoming the recurrence in 5-FU-treated CRC (Fig. 6j).

## Discussion

Since the 1950s, 5-FU-based chemotherapy has remained the mainstay of therapy for patients with CRC. Although combinations of 5-FU and leucovorin with oxaliplatin or irinotecan have conferred survival benefits greater than the use of 5-FU alone, nearly half of CRCs are resistant to 5-FU-based chemotherapies. Therefore, extensive studies have focused on the characterization of the biological factors involved in mediating resistance to 5-FU-based therapy or to the identification of predictive biomarkers to define those patients who are most likely to benefit from 5-FU-based chemotherapy[5,10,26,38]. Even in patients with an initial response to this therapy, recurrence after chemotherapy still limits the clinical outcomes of the patients, indicating an urgent need for development of targeted therapies that halt chemotherapy-induced recurrence[4].

P53, one of the most important tumor suppressors in CRC, dictates the sensitivity to 5-FU-based therapies[22–24]. In addition to its well-known transcriptional activities controlling the expression of a variety of genes important in anti-cancer effects, its crosstalk with other signaling pathways in various cellular processes has been identified[28,29]. Recent studies demonstrate unanticipated roles of p53 in mouse ESCs. P53 directly coordinates WNT and Smad pathways and plays essential roles in differentiation of pluripotent ESCs[28,29]. Moreover, upon DNA damage, p53 activates WNT signaling via WNT ligand transcriptional induction, showing the two faces of p53 regulating both the pro- and anti-survival of the ESCs[28,29].

In this study, we identified the involvement of p53 in 5-FU-induced CSC activation. Intriguingly, upon treatment with 5-FU, p53 activated the WNT/β-catenin signaling pathway via transcriptional induction of *WNT3*, leading to activation and enrichment of CSCs in the tumor. No significant difference in the basal β-catenin level among *p53$^{+/+}$*, *p53$^{-/-}$*, and *p53$^{R248W/-}$* isogenic cells indicates that in addition to the *WNT3* induction by p53, other mechanisms mediated by p53 may also be involved in the negative regulation of the basal β-catenin level, as shown by previous studies[39,40]. Therefore, p53 may play both negative and positive roles in the regulation of WNT/β-catenin signaling in the intact status; however, upon 5-FU treatment, p53 activates the WNT/β-

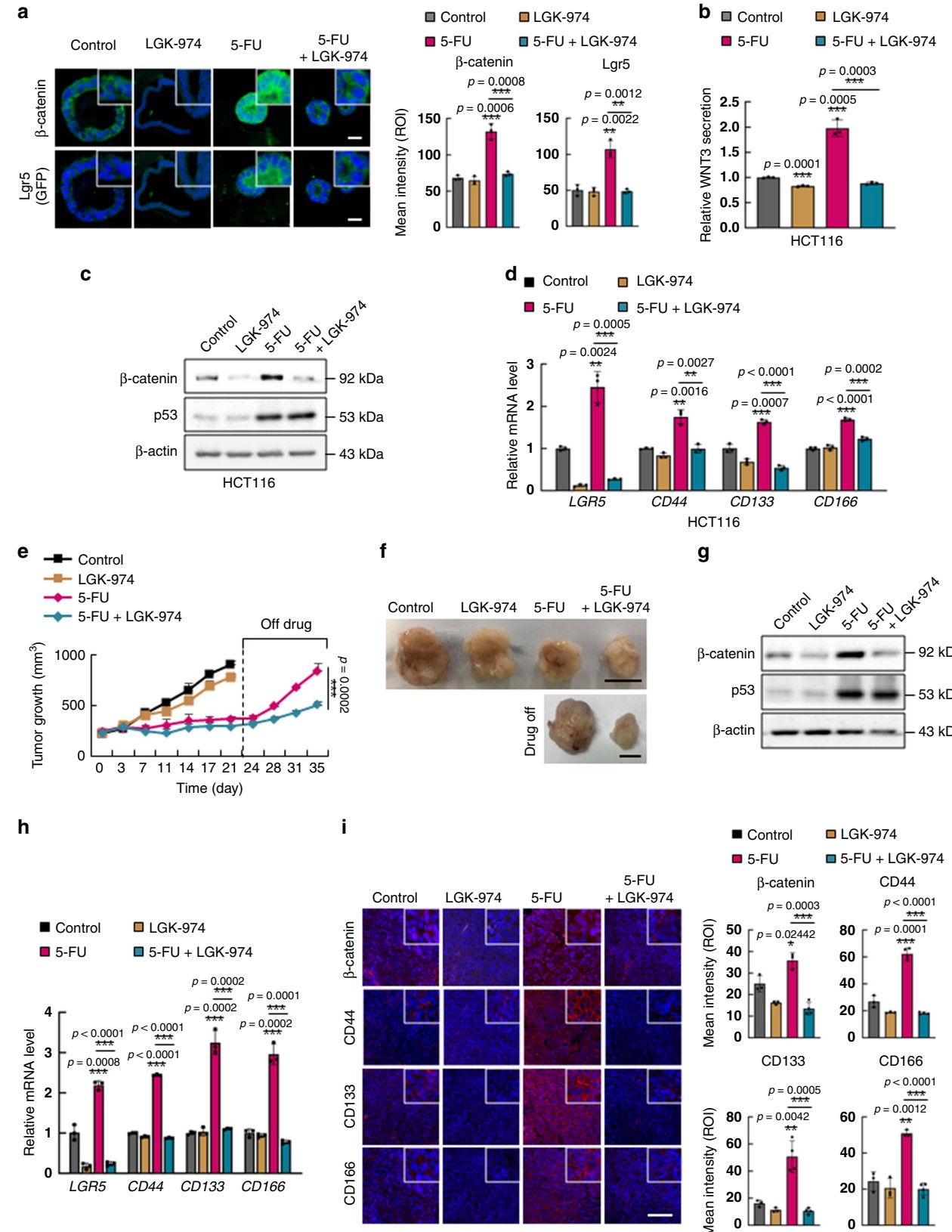

catenin pathway via induction of *WNT3*, rendering p53 as a two-faced regulator exerting anti-cancer and CSC-activating roles.

Our findings reveal that p53 may contribute to clinical outcomes of 5-FU treatment in CRC harboring wild-type *p53* via its dual roles in anti-cancer effects and CSC-inducing effects. Furthermore, 5-FU-induced CSC activation by p53-mediated

activation of WNT/β-catenin signaling suggests that targeting of WNT/β-catenin signaling is the key to increase the clinical benefit of 5-FU. Therefore, combinatory treatment of the 5FU-based therapy with a WNT/β-catenin signaling inhibitor may be beneficial for suppression of CSCs and therefore minimization of recurrence in CRC. In PDTOs and in vivo xenograft models, 5-

**Fig. 5 WNT inhibition suppresses 5-FU-induced enrichment of CSCs and tumor regrowth. a** Confocal images of β-catenin (green) and Lgr5 (GFP) (green) in $Apc^{Min/+}/Lgr5^{EGFP}$ intestinal tumor organoids co-treated with 5-FU (1.5 μg/ml, 48 h) and LGK-974 (5 μM, 48 h) (left panel) and quantification of the mean intensities (right panel). Scale bar = 50 μm. ($n = 3$ biologically independent samples per group). **b** Measurement of WNT3 secretion by ELISA analyses using 12 h cultured medium of HCT116 cells after co-treatment with 5-FU and LGK-974. ($n = 3$ biologically independent samples per group). **c** Immunoblots of indicated proteins in HCT116 cells co-treated with 5-FU and LGK-974. **d** Relative mRNA levels of CSC markers *LGR5*, *CD44*, *CD133*, and *CD166* in HCT116 cells co-treated with 5-FU and LGK-974. ($n = 3$ biologically independent samples per group). **e–i** HCT116 cells were xenografted in mice and tumors were extracted and analyzed after co-treatment with 5-FU (25 mg/kg) and LGK-974 (5 mg/kg) daily for 21 days followed by discontinuation of treatment for 14 days. **e** Growth of tumors were measured at the indicated days after treatments. ($n = 5$ in D0, D3, D7, D11, D14, D17, and D21 biologically independent animals per group and $n = 4$ in D24, D28, D31 biologically independent animals per group). **f** Gross images of tumors. Scale bar = 10 mm. **g** Immunoblots of indicated proteins in isolated tumors after indicated treatment for 21 days. **h** Relative mRNA levels of *LGR5*, *CD44*, *CD133*, and *CD166* in isolated tumors after indicated treatment for 21 days. ($n = 3$ biologically independent samples per group). **i** Immunohistochemistry analysis of indicated proteins in sections of isolated tumors after treatment for 21 days and quantification of mean intensities. Scale bar = 100 μm. The mean intensities were measured by Zen software 3.1. (β-catenin $n = 3$ in Control, LGK-974 and 5-FU, and $n = 4$ in 5-FU+LGK-974 group, CD44 $n = 3$ in Control, LGK-974, and $n = 4$ in 5-FU and 5-FU+LGK-974 group, CD133 $n = 3$ in Control, LGK-974, and $n = 4$ in 5-FU and 5-FU+LGK-974 group, CD166 $n = 3$ in Control, LGK-974 and 5-FU, and $n = 4$ in 5-FU + LGK-974 group). Data are mean ± s.d., two-sided Student's *t*-test, **$p < 0.01$, ***$p < 0.001$. Source data are provided as a Source data file.

FU treatment activated WNT/β-catenin signaling and enriched CSCs in tumors, and regrowth of the residual tumors were observed following discontinuation of treatment. Importantly, combinatorial treatment with a WNT inhibitor and 5-FU effectively suppressed CSC enrichment in the remaining tumors, effectively repressing the tumor regrowth.

As an essential niche factor for the maintenance of ISCs, WNT ligand production by the epithelium and the mesenchyme is indispensable for normal intestinal homeostasis[15]. Despite the similar characteristics of ISCs with CSCs, the importance of WNT ligand in CSC regulation in CRC has not been identified. Although the details of the involvement of wildtype and mutant APC and functional redundancy of APC2 in the responsiveness of down-stream signaling to WNT ligand in CRCs harboring APC loss of function remain unknown, our findings revealed a critical role for WNT ligand in the activation and enrichment of CSCs in the 5-FU-treated CRC tumors. The combinatorial treatment with WNT inhibitor and 5-FU reduced the clonal expansion of CSCs, minimizing recurrence as expansion of CSCs gives rise to the heterogeneous population and results in recurrence. Our findings identified critical roles of wildtype p53 in activation of WNT/β-catenin signaling and CSCs upon 5-FU treatment, suggesting that targeting the WNT/β-catenin signaling with 5-FU treatment could be a potential therapeutic strategy to overcome the recurrence after 5-FU treatment. However, our proposed model do not preclude the WNT/β-catenin signaling activation by mutant p53 in CRC. Although the underlying mechanisms are different, both mutant p53 and wildtype p53 may result in WNT/β-catenin signaling activation. Thus, it is possible that inhibitors of WNT/β-catenin signaling, though probably not LGK-974 but rather cell-intrinsic inhibitors, such as the downstream WNT/β-catenin signaling inhibitors that showed effectiveness in CSC suppression such as KYA1797K[41], might be beneficial to mutant p53 harboring color-ectal cancer patients, while both WNT ligand and downstream WNT/β-catenin signaling inhibitors may be useful to prevent relapse after 5-FU treatment in WT p53 harboring patients.

## Methods

**Cell cultures, and reagents.** Human CRC cells (LoVo, HCT116, RKO, SW48, DLD-1, SW480, SW620, and WiDr), were purchased from the American Type Culture Collection (Manassas, VA). *P53* wild-type ($p53^{+/+}$), null ($p53^{-/-}$) and mutant ($p53^{R248W/-}$) and *p21* wild-type ($p21^{+/+}$) and null ($p21^{-/-}$) isogenic HCT 116 cell lines were provided by B. Vogelstein (John Hopkins oncology center, MD). All cell lines were authenticated using short tandem repeat profiling (Cosmogen-etech, Korea) and were maintained in RPMI1640 (Gibco, CA) or DMEM (Gibco) supplemented with 10% fetal bovine serum (Gibco). RITA (Santa Cruz Bio-technology, CA), pifithrin-α-p-nitro-cyclic (pifithrin-α) (Santa Cruz Biotechnol-ogy), LGK-974 (Selleckchem, TX) were dissolved in dimethyl sulfoxide (Sigma-Aldrich, MO) for the in vitro studies.

**Establishment of 5-FU-resistant cells.** HCT116, SW480, and HT29 cells were exposed to an initial 5-FU concentration of 0.1 μg/ml in DMEM plus 10% FBS. The surviving population of cells was grown to 80% confluence and passaged twice over 9 days to ensure viability. 5-FU-R cells were established after sequential treatments of 5-FU with increasing concentration of 5-FU as 0.25, 0.5, and 1.5 μg/ml, and the maximal concentration was used 6 times until cells could live in the 5-FU maximal concentration for more than 3 days. The chronic resistances to 5-FU were con-firmed by MTT assay at the 5-FU treatment dose of 1.5 μg/ml for 72 h. Control parental cells were passaged in parallel.

**Spheroid culture.** In all, $1 \times 10^4$ cells/ml were seeded with serum-free medium containing DMEM/F12 (Invitrogen, CA) supplemented with B27 (Invitrogen) and 20 ng/ml EGF and 10 ng/ml bFGF (Peprotech, NJ) in ultra-low attachment plates (Corning, NY). Experimental procedures were performed after 5 days of spheroid-forming culture. Spheroid-forming ability was measured by the size of the spher-oids using Image J software V1.46.

**Immunoblotting.** After ice-cold PBS washing, cells were lysed using radio-immunoprecipitation assay (RIPA) buffer [150 mM NaCl, 10 mM Tris pH 7.2, 0.1% sodium dodecyl sulfate, 1% Triton X-100, 1% sodium deoxycholate, and 5 mM ethylenediaminetetraacetic acid]. Samples of mouse tumor tissues were stored in liquid nitrogen andhomogenized with RIPA buffer. Proteins were separated on a 10–12% sodium dodecyl sulfate polyacrylamide gel and transferred to a nitro-cellulose membrane (Whatman, Maidstone, UK). Immunoblotting was performed according to standard protocol with the following primary antibodies: anti-p53 (Santa Cruz Biotechnology, sc-126; 1:1000), anti-p21 (Santa Cruz Biotechnology, sc-6246; 1:1000), anti-β-catenin (Santa Cruz Biotechnology, sc-7199; 1:3000), anti-active β-catenin (Sigma-Aldrich, 05-665; 1:1000), anti-WNT3 (Abcam, Cambridge, UK, ab32249; 1:1000), anti-p-LRP6 (Cell Signaling Technology, MA, #2568; 1:1000), anti-β-actin (Santa Cruz Biotechnology, sc-47778; 1:5000), horseradish peroxidase-conjugated anti-mouse (Cell Signaling Technology, #7076; 1:3000) or anti-rabbit (Bio-Rad, CA, #1706515; 1:3000) secondary antibodies were used. Protein band detection was performed using a luminescent image analyzer (LAS-4000, Fuji Film, Japan; ChemiDoc XRS+System, Bio-Rad, CA) and analyzed by using MultiGauge software V3.0 (Fuji Film) and Image Lab software V6.0.1 (Bio-Rad). Original scans of the most important blots are supplied in the Source Data.

**Reverse transcription and quantitative real-time PCR.** Total RNA was isolated using Trizol® reagent (Invitrogen, CA) following the manufacturer's instructions. The cDNA was synthesized from total RNA (2 μg) using 200 units of M-MLV reverse transcriptase (Invitrogen) in a 20 μl reaction mixture carried out at 42 °C for 1 h. PCR amplification performed using the SYBR PCR master mix (Qiagen, Hiden, Germany), and the PCR-amplified gene products were analyzed. Levels of mRNA expression were quantified after normalization to *β-actin* endo-genous control using the ΔCT (difference between cycle thresholds) method. Three biological replicates were performed.

**Enzyme-linked immunosorbent assay (ELISA).** The measurements of WNT3 secretion were performed by the WNT3 ELISA kit (#MBS919345, MyBiosource, Sandiego) according to the manufacturer's instruction. Briefly, 12 h cultured media were incubated in the WNT3 antibody-bound 96 wells for 2 h at 37 °C. The wells were incubated with biotin-conjugated antibody for 1 h at 37 °C. Next, the wells were incubated with Avidin-HRP antibody for 1 h at 37 °C. The wells were incubated with TMB substrate for 15 min at 37 °C. The amount of WNT3 in cultured media was measured by the FLUOstar Optima instrument (BMG Labtech, Ortenberg, Germany).

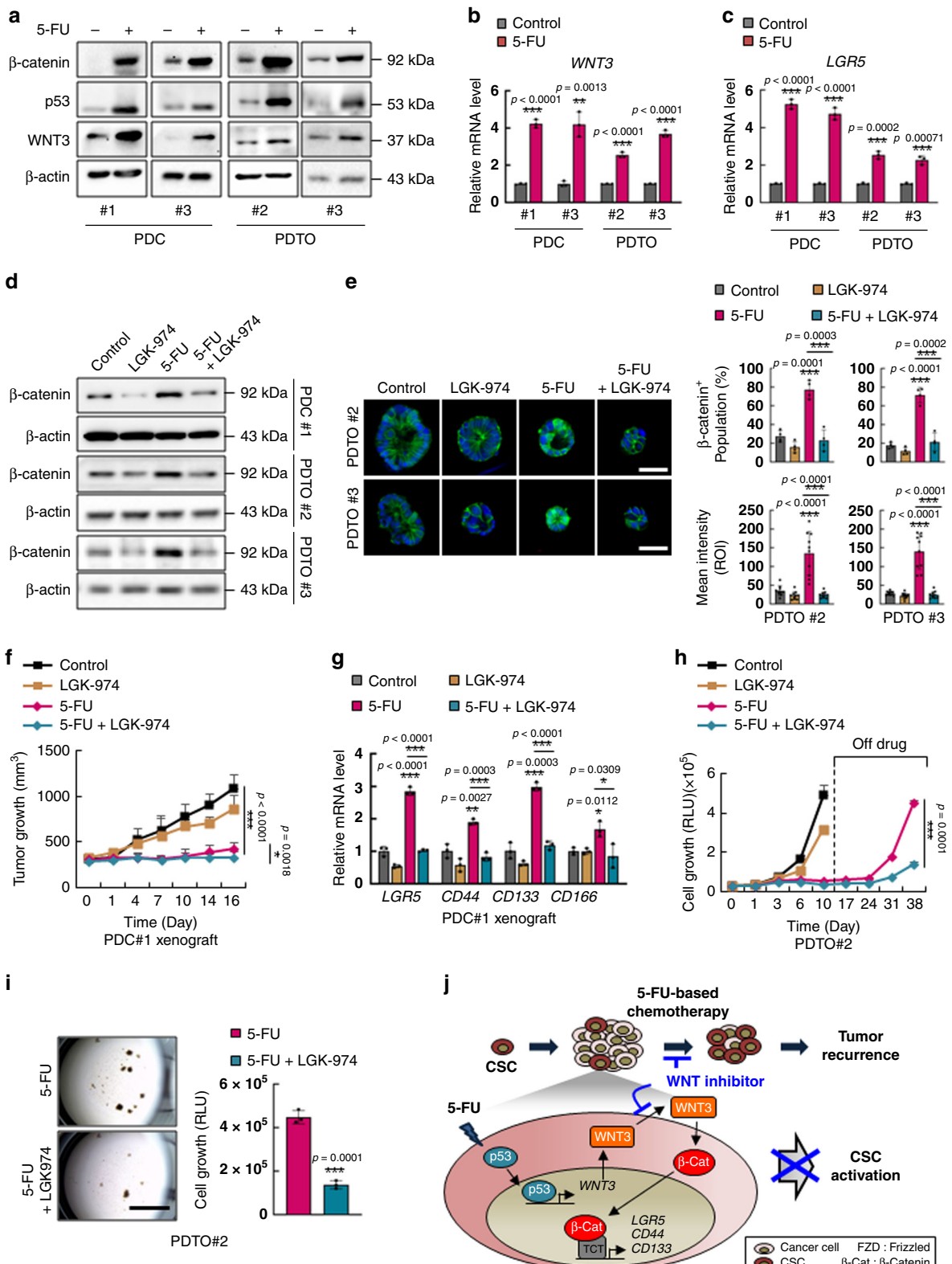

**Genetic animal study**. All animal experiments were conducted in accordance with the guidelines of Korean Food and Drug Administration. Protocols were reviewed and approved by the Institutional Animal Care and Use Committee of Yonsei University. The C57BL/6J-$Apc^{Min//+}$ ($Apc^{Min/+}$) mice, B6.129S-$Kras^{tm3Tyj}$ ($Kras^{G12D}LA2$), and B6N.129P2-$Axin2^{tm1Wbm}$/J ($Axin2^{LACZ}$) were obtained from Jackson Laboratory (Bar Harbor, ME). Lgr5$^{EGFP-IRES-creERT2}$ ($Lgr5^{EGFP}$) mice were provided by Young Yun Kong (Seoul National University, Korea) to generate $Apc^{Min/+}$/$Lgr5^{EGFP}$, $Apc^{Min/+}$/$Kras^{G12D}LA2$/$Lgr5^{EGFP}$, and $Apc^{Min/+}$/$Axin2^{LACZ}$. Mouse genotyping was performed using genomic DNA extracted from the tail. To

control for genetic background effects, sex-matched littermates were used as controls. For in vivo experiments, 12-week-old $Apc^{Min/+}$/ $Lgr5^{EGFP}$ ($n = 4$), $Apc^{Min/+}$/$Kras^{G12D}LA2$ ($n = 3$) mice were intraperitoneally injected with 5-FU (25 mg/kg) 3 days per week for 3 weeks.

**Xenograft study**. Athymic $nu/nu$ mice and SCID/g (NOD-Prkdc $^{scid\ tm1Baek}$) mice (GemBiosciences, Chungbuk, Korea) were injected subcutaneously in dorsal flank with HCT116 ($5 \times 10^6$ cells/mice) and PDC#1 ($1 \times 10^7$ cells/mice), respectively, in

**Fig. 6 WNT inhibitor effectively suppresses 5-FU-induced enrichment of CSCs and tumor regrowth in CRC patients. a** Immunoblots of indicated proteins in PDCs and PDTOs with or without 5-FU treatment (1.5 μg/ml, 48 h). **b, c** Relative mRNA levels of *WNT3* (**b**) and *LGR5* (**c**) in PDCs and PDTOs with or without 5-FU treatment. (*n* = 3 biologically independent samples per group). **d** Immunoblots of indicated proteins in PDCs and PDTOs co-treated with 5-FU and LGK-974 (5 μM, 48 h). **e** Confocal images of β-catenin (green) in PDTOs co-treated with 5-FU and LGK-974 (left panel) and quantification of the percentage of β-catenin⁺ population among the cells in each tumor organoid (upper right panel) (*n* = 4 biologically independent samples per group) and the mean intensities (lower right panel). (*n* = 10 biologically independent samples per group) Scale bars = 50 μm. **f, g** PDC#1 xenograft mice were treated with 5-FU (25 mg/kg) alone or co-treated with 5-FU (25 mg/kg) and LGK-974 (5 mg/kg) for 16 days and tumors were extracted and analyzed. **f** Tumor growth in mice during the treatments is shown. (Control *n* = 8, LGK-974 *n* = 6, 5-FU *n* = 8, 5-FU+LGK-974 *n* = 8 biologically independent animals) **g** Relative mRNA levels of *LGR5*, *CD44*, *CD133*, and *CD166* in isolated tumors after treatment. (*n* = 3 biologically independent samples per group) **h, i** Tumor organoids derived from patient#2 were treated with 5-FU (1.5 μg/ml) alone or co-treated with 5-FU and LGK-974 daily for 10 days followed by discontinuation of treatment for 28 days. **h** Growth of PDTO#2 at indicated days. (*n* = 10 in D0, *n* = 3 in D1, *n* = 7 in D3, D6, D10, *n* = 4 in D17, *n* = 3 in D24, D31 and D38 biologically independent samples per group). **i** Bright field image of PDTO#2 at 28 days after discontinuation of treatment of 5-FU alone or co-treatment of 5-FU and LGK-974 (left panel) and measurement of cell growth in PDTO#2 by cell titer assay (right panel) (*n* = 3 biologically independent samples per group). Scale bars = 4 mm. **j** Schema depicting the mechanism by which 5-FU induces activation and enrichment of CSCs via p53. The mean intensities were measured by Zen software 3.1. Data are mean ± s.d., two-sided Student's *t*-test, *$p < 0.05$, **$p < 0.01$, ***$p < 0.001$. Source data are provided as a Source data file.

200 μl of PBS: Matrigel® (BD Biosciences, CA; 1:1). When the mean tumor size reached between 100 and 200 mm³, the mice were randomly divided into four groups; Vehicle, LGK-974, 5-FU, 5-FU+LGK-974 groups. LGK-974 was orally injected every day at the dose of 5 mg/kg and 5-FU was intraperitoneally injected three times a week at the dose of 25 mg/kg.

**Immunohistochemistry**. 4-μm paraffin-embedded tissue samples were incubated with 10 mM sodium citrate buffer (pH 6.0) and autoclaved for 15 min for antigen retrieval. The samples were then blocked with PBS-based mixture of 5% bovine serum albumin (BSA; Affymetrix, CA) and 1% normal goat serum (NGS, Vector Laboratories, CA) for 30 min. After Blocking, sections were incubated with GFP (Santa Cruz Biotechnology, sc53882, 1:100), β-catenin (Abcam, ab2365, 1:100), CD44 (Proteintech, #15675-1-AP, 1:100), CD133 (Miltenyi Biotec, #130-108-062, 1:50), and CD166 (Abbiotec, #251619, 1:100) primary antibody overnight at 4 °C, followed by incubation with anti-mouse Alexa Fluor 488 (Life Technologies, CA, A11008; 1:500) or anti-rabbit Alex Fluor 555 (Life Technologies, A21428; 1:500) secondary antibodies for 1 h at room temperature. The samples were then counterstained with 4′, 6-diamidino-2-phenylindole (DAPI; Sigma-Aldrich) and mounted in Gel/Mount media (Biomeda Corporation, CA). All incubations were conducted in dark, wet chambers. The fluorescence signal was visualized using a confocal microscope (LSM510; Carl Zeiss, Germany) and analyzed by using Zen software V3.1 (Carl Zeiss).

**Mouse tumor organoid experiments**. Small intestinal tumors of *Apc^{Min/+}/ Lgr5^{EGFP}*, *Apc^{Min/+}/ Kras^{G12D}LA2/ Lgr5^{EGFP}*, and *Apc^{Min/+}/Axin2^{LACZ}* mice were isolated and washed with ice-cold PBS, and single cells isolated from tumors were collected using 0.25% trypsin containing 10 μM LY27632 (Sigma-Aldrich) and 100 μg/ml Primocin™ (Invivogen, CA) for 30 min. After incubation, B27 (Sigma-Aldrich) was added and the cells were filtered through 100 μm and 40 μm cell strainers (BD Biosciences) to collect single cells, and then 250 cells were mixed with 25 μl of growth factor reduced Matrigel® per well. After the gels were solidified, N2 medium containing 10% R-spondin-1 CM, 100 μg/ml of EGF, 100 μg/ml of noggin The growth medium was refreshed every 2 days and the cells were passaged by mechanical disruption every 10–14 days with a 1:5 split ratio. The measurements of tumor organoid growth were performed by the Cell Titer-Glo® Luminescent Cell Viability Assay (Promega) according to the manufacturer's instructions. Luminescence was measured by the FLUOstar Optima instrument (BMG Labtech).

**CRC patient tumor-derived cultures**. CRC PDTOs and PDCs[33] were used after receiving patient-informed consent and approval from the Institutional Review Board of Yonsei Severance hospital and Asan Medical Center, respectively. *APC^{KO}/ p53^{WT}* and *APC^{KO}/p53^{KO}* human colon organoids were established as described in detail by Drost et al.[31] and was approved by the ethical committees of the Diakonessen Hospital Utrecht. Written informed consent for the study was obtained from patients. All procedures performed in studies involving human participants were conducted in accordance with International Ethical Guidelines for Biomedical Research Involving Human Subjects. For PDTO, 250 cells per 25 μl of growth factor reduced Matrigel® (BD Bioscience) were mixed and seeded, and N2 medium containing 10% R-spondin-1 CM, 100 μg/ml noggin (Peprotech, NJ), 1.25 mM N-acetyl cysteine (Sigma-Aldrich), 10 mM nicotinamide (Sigma-Aldrich), 50 ng/ml EGF (Peprotech) and bFGF (Peprotech), 10 nM gastrin (Sigma-Aldrich), 500 nM A83-01 (Sigma-Aldrich), and 3 μM SB202190 (Sigma-Aldrich) were added. Inhibitors such as 500 nM A83-01 (Sigma-Aldrich), 3 μM SB202190 (Sigma-Aldrich) were excluded for the investigation of 5-FU effects. The growth medium was refreshed every 2 days and the cells were passaged by mechanical disruption every

10–14 days with a 1:5 split ratio. The measurements of tumor organoid growth were performed by the Cell Titer-Glo® Luminescent Cell Viability Assay according to the manufacturer's instructions. Luminescence was measured by the FLUOstar Optima instrument.

**Tumor organoid immunocytochemistry**. The cells were fixed with 5% formalin for 30 min, permeabilized with 0.2% Triton X-100 for 30 min, and pre-blocked with PBS containing 5% BSA and 1% NGS for 1 h. The cells were then incubated with the GFP (Santa Cruz Biotechnology, sc53882, 1:100), β-catenin (Abcam, ab2365, 1:100), p53 (Santa Cruz Biotechnology, sc-126; 1:100), and WNT3 (Abcam, Cambridge, UK, ab32249; 1:100) primary antibody overnight at 4 °C, followed by incubation with Alexa Fluor 488 or Alex Fluor 555 secondary antibodies for 1 h at 4 °C, and then counterstained with DAPI for 10 min at room temperature. After incubation, the cells were mounted in Gel/Mount media (Biomeda Corporation, CA). The fluorescence signal was visualized using a confocal microscope (LSM700) at excitation wavelengths of 488 nm (Alexa Fluor 488), 543 nm (Alexa Fluor 555), and 405 nm (DAPI).

**Statistics and reproducibility**. All data are represented as the mean ± standard deviation of at least three independent experiments. The statistical significance of differences was assessed by the Student's *t*-test using Prism software V8.4.3 (Graphpad, CA). *P* values < 0.05 were considered statistically significant. Significance was denoted as n.s. not significant, *$p < 0.05$, **$p < 0.01$, and ***$p < 0.001$. All data shown were obtained from at least three biological independent experiments.

**Reporting summary**. Further information on research design is available in the Nature Research Reporting Summary linked to this article.

## Data availability
Disease-free survival rate data in patients with differential stages after 5-FU-based treatment discussed in this publication have been obtained from a publicly available source in NCBI's Gene Expression Omnibus[42] and is accessible through GEO series accession number GSE14333. The source data underlying Figs. 1a–f, 2b, d, f, 3b, g, h, 4a–i, 5a, b, d, e, h, i, 6b, c, e–i and Supplementary Figs. 1, 3a, c, 4c, 5a, b, d, f, g, i, 6b, e, 7b, c, e, g, 8a, c, e, and g are provided as a Source data file. All the other data supporting the findings of this study are available within the article and its supplementary information files and from the corresponding author upon reasonable request. A reporting summary for this article is available as a Supplementary Information file.

## Code availability
The computer codes that support the plots within this paper are available from the corresponding author upon reasonable request.

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

## Acknowledgements

This work was supported by the National Research Foundation of Korea (NRF) grant funded by the Korean Government (MSIP) (grants 2019R1A2C3002751, 2018R1D1A1B07050189).

## Author contributions

Y.H.C., E.J.R., J.S.Y., and J.C.P performed the experiments. T.M. and H.C. shared $APC^{KO}/p53^{WT}$ and $APC^{KO}/p53^{KO}$ human colon organoids. D.W.K shared PDCs. T.I.K shared CRC patient tumors. K.Y.C. supervised the study. Y.H.C., E.J.R. and K.Y.C. performed data analysis and wrote the paper.

## Competing interests

The authors declare no competing interests.
