## [Peer Review File · Nature Communications]

Reviewers' comments:

Reviewer #1 (Remarks to the Author); expert in 5-FU resistance:

Using multiple in vitro and in vivo model system authors wanted to show that the presence of cancer stem cells in the bulk of cancer cells is the main reason for 5 FU resistance in chemotherapy. They also prove that WNT inhibitor along with 5FU increases the sensitivity of the treatment. They conclude that p53 is the critical mediator of 5-FU-induced CSC activation via the WNT/ β -catenin signaling pathway.

I have some concern of the MS

- 1) It will be better if authors make 5-FU resistance cell lines using different APC and P53 CRC cells and then carried out the experiment instead of 5FU treatment.
- 2) They need to show that enrichment of CSC happen due to 5 fu resistance, for that they need to make the CSC cells from 5-fu resistance cells and then characterized.
- 3) For the mechanistic study the development of CSC model from 5-FU resistance cell will be better than transient treatment with 5FU.
- 3) Why they only focus on WNT signaling, not others potential CSC signaling
- 4) Why they use 5Fu in combination with WNT inhibitor to increase the sensitivity of 5FU resistance. if the tumor already 5Fu resistance then how with 5FU it will be sensitive.

Reviewer #2 (Remarks to the Author); expert in Wnt signalling:

In this report, the authors investigated the effects of a chemo agent 5-FU on colorectal cancer stem cells (CSCs) and asserted that 5-FU stimulates production of Wnt3 in p53 WT colorectal cancer (CRC) cells, which in turn via β -catenin drives Lgr5 and other CSC stem cell marker expression and renders the tumor resistance to 5-FU, leading to CRC recurrence. While there are some intriguing findings revealed by the study, there are many unanswered questions that would mis-lead readers or obscure the real mechanisms.

- 1) It seems reasonably convincing that 5-FU drives Wnt3 expression in p53 WT cells. However, it becomes murky how or whether 5-FU can drive β -catenin increase or nature/mechanism of this increase. It has been shown by many that Wnt does not drive substantial β -catenin stabilization in APC-loss cells including CRC cells. Would the conclusion of this study be only applied to CRCs with functional APC? It seems that 5-FU increases total β -catenin in APC loss cells (LoVo). Are there alternative mechanisms for the increases in β -catenin levels (see #2)
- 2) The Wnt- β -catenin pathway regulates the stability of β -catenin. Total cellular β -catenin can be regulated not only by Wnt. Thus, cytosolic or nuclear β -catenin contents have to be examined if Wnt signaling is claimed. Along the same line, β -catenin mRNA and cadherin pathway need to be examined to determine if alternative mechanisms are involved.
- 3) What exactly the effects of 5-FU on Lgr5+ cells, converting non-Lgr5 into Lgr5+ cells or Lgr5+ cells are more resistant to 5-FU. Careful analysis of proliferation and apoptosis of Lgr5+ cells probably by flow cytometry, together with some kind lineage tracing means for Lrg5 conversion, is needed to address this question.
- 4) Imaging-based quantifications are not reliable, unless ratio imaging is used. Western should be used in all cases.
- 5) The increase in mRNA of Wnt3 in HCT116 does not seem to be pronounced. Can the effect of 5-FU be recapitulated by adding exogenous Wnt? Comprehensive analysis of the gene expression of 5-FU-treated cells may be needed to gain a more complete picture. Wnt3 may be only a small part of the truth.
- 6) Can the LGK-974 effect be reproduced in organoids derived from single APC-null stem cells? Similarly, can the data in Figure 5 using HCT116 be reproduced with APC-loss cells? Additional means to inhibit Wnt signaling is needed as LGK may have off-target effects.
- 7) Why does β -catenin staining show punctate patterns in Fig. 6e rather than uniform as other staining in the study? Does it mean that cells in the organoids are heterogenous? The changes in

β -catenin levels in fig 6d are less pronounced. It is hard to believe that the combo effects shown in Fig. 6h,I are exclusively due to these small differences in β -catenin levels, especially we do not even know what kind of β -catenin it is. The real mechanism could be more complex than what is asserted here.

Reviewer #3 (Remarks to the Author); expert in CRC organoids and cancer stem cells:

This is an interesting paper in which Cho and colleagues suggest a potential mechanism of acquired 5FU resistance via p53 mediated Wnt activation. Overall the data is of a good quality and clearly laid out and the results point to a potential way to overcome 5-FU resistance in some colorectal cancer patients. I have a number of comments on the study and some experiments I think are important for supporting the conclusions of the work.

Major points

- 1) The overall conclusions of the study seem counter intuitive. In the introduction the authors outline some known functions of P53 in response to 5FU. All of these are tumour suppressive but this study suggests the opposite, that P53 is an important mediator of disease relapse. Due to the apparent contradiction, more supporting evidence from human patient cohorts is needed. For example, are patients with wild type P53 more likely to relapse? Do they have shorter survival times than P53 mutant patients? What does Wnt signalling look like in these patients?
- 2) The authors demonstrate that following 5FU treatment, organoids exhibit increased stem cell marker expression. What about stem cell function? Clonogenicity assays should be carried out to assess if these are functional stem cells.
- 3) What does 5FU do to the organoids and why does Wnt signalling protect Lgr5+ stem cells? The authors should investigate apoptosis and proliferation in the 5FU treated organoids, in particular, are Lgr5+ stem cells more resistant to the effects?
- 4) Related to this, is Wnt3 sufficient to drive these phenotypes? Does Wnt3 of organoids / cell lines lead to the same Bcat stabilisation and stem cell marker increases? And does this impact on 5FU treatment response?

Minor points

- 1) What is the dataset used for Fig S1? Is P53 status known and does this impact on survival?
- 2) The IHC images for Fig 2 are poor quality and better resolution images are needed.
- 3) The cell line data in Fig 3a is not clear to me. It looks like the P53 mut cell lines have higher basal B-cat than the p53 wt. Does this suggest a p53 independent mechanism of increasing Wnt signalling in these cells lines?

Reviewer #4 (Remarks to the Author); expert in p53 and cancer stem cells:

In the current study, Cho and colleagues aim to discover what governs tumor recurrence after 5-FU treatment in colorectal cancer (CRC) patients. They hypothesize that tumors recur after 5-FU treatment due to expansion of cancer stem cells (CSCs) population, and further aim to elucidate the molecular mechanisms that govern this expansion and tumor recurrence. They begin by demonstrating that indeed 5-FU treatment, while inhibiting tumor cell growth, enriches the CSC population in CRC organoids and murine tumors. They further demonstrate that this enrichment, which is caused by 5-FU treatment, is associated with increased beta-catenin expression, which is a hallmark of WNT pathway activation. Next, they show that the activation of beta catenin after 5-FU treatment is caused by p53 activation. They next demonstrate that p53 upregulates WNT3, which is associated with beta catenin upregulation. In addition, the authors demonstrate that the activation of the WNT pathway after 5-FU treatment can be inhibited by a WNT secretion inhibitor, which also inhibits the recurrence of tumors in vivo via inhibiting the activation of CSCs. Finally, the authors demonstrate the aforementioned findings also in patient-derived cell-lines and organoids, thus demonstrating potential clinical relevance.

These findings are indeed very intriguing, and demonstrate how WT p53 activation, at least in colon cancer, can have negative implications on eventual tumor outcome, and suggest a potential therapeutic strategy to overcome them. This manuscript may be indeed of high interest to the scientific and medical community, provided that the following issues are addressed:

1. Figure 3 – In panel A, it is worthy to notice that although beta catenin is indeed upregulated (compared to the basal level) after 5-FU treatment only in WT p53 expressing cells, it is noticeable that in mutant p53 expressing cells, beta catenin is already activated in basal level, although indeed there is no further activation after 5-FU treatment. As for comparison between the effects of different p53 status on an isogenic cellular background of HCT116, there seems to be a discrepancy between the results shown in panel C and panel E. While panel C shows that in HCT116 p53^{-/-} there is upregulation of beta catenin already in basal level, without 5-FU treatment, it does not seem so in the experiment depicted in panel E. This aspect, although not pertaining to 5-FU response directly, is important in terms of p53 regulation of the WNT pathway, which is further important to the activation of CSCs which is a major part of this work. The authors do not address this point neither in the results section nor in the discussion section. Indeed, WNT3, which activates the WNT pathways in stem cells in a non-cell autonomous manner, was shown to be a WT p53 target both by the authors and by a previous publication that the authors reference in their manuscript (Wang et al (1)). However, several other publications show that WT p53 is a negative regulator of the WNT pathway, in a cell autonomous manner and via different mechanisms (2–5), and that mutant p53 may even upregulate beta-catenin (6), which is in line with the aforementioned results shown in panels A and C. Therefore, I think several issues are needed to be addressed in that regard:

A. In panel 3C – were the HCT116^{-/-} ran on the same gel as HCT116^{+/+}? If not, please repeat the experiments and run on the same gel so beta catenin levels could be compared. Of note, it may be of value, if possible and obtainable, to include HCT116 p53R248W^{-/-} mutant p53-expressing sub-line, which was also generated by prof. Bert Vogelstein's laboratory, in this experiment.

B. How do the authors explain the discrepancy between the experiments in panel C and E? If one of the experiments seems to be technically incorrect, please exclude it from the manuscript.

C. Please address the aforementioned known relations between the WNT signaling and p53 in the results and/or in the discussion section/s, and how do you interpret your results in light of this aforementioned literature.

D. According to the current authors' interpretation, only patients that harbor WT p53 may benefit from a combined 5-FU and WNT inhibitor treatment. However, as mentioned above, mutant p53 expressing cells may also exhibit increased WNT activation, which may lead to increased CSC population (although this was not addressed in the current manuscript), independently of 5-FU treatment. Therefore, do the authors indeed expect that the combined treatment will be applicable only for patients harboring tumor with intact WT p53? Please address this issue in the discussion section as well.

2. It seems to me that a more accurate way to quantify WNT pathway activation, which leads to expansion of CSCs (e.g. in figure 2B+D, figure 3b, 3h-I etc.), would be by assessing percentage of (nuclear?) beta catenin positive cells (similar to what the authors did for LGR5 in panel 1d), rather than the mean intensity of the protein levels, which could reflect a stronger activation of beta catenin in the same cells.

3. Figure 4i – It would be beneficial to assess secreted WNT3 protein levels (e.g. by ELISA) and not only WNT3 levels by IHC, which could represent WNT3 which is accumulated the cells that was not secreted. This is also relevant for the experiment described in figure 5A-C, in order to assess whether LGK-974 affected the levels of secreted WNT3, therefore demonstrating that the associated phenotypes are due to depletion of secreted WNT3.

4. For the in-vivo experiment described in figure 5d-h – an assessment of p53 activation is missing to demonstrate the link between p53 and the induction of WNT signaling in-vivo. Please assess p53 activation by WB and/or IHC as depicted in panels 5f and 5h, respectively.

5. Figure 6 – please include at least one patient-derived cell-line and organoid with mutated p53 as a negative control for p53, WNT3 and LGR5 induction after 5-FU, at least for the in-vitro assays.

6. Based on the authors' results, especially those that demonstrate activation of beta catenin and CSC expansion after RITA treatment, it could be speculated that any compound or drug that activates p53, such as irinotecan or oxaliplatin, which are also used in CRC, could result in the same CSC enrichment and tumor recurrence. Do the authors expect this to be true? Perhaps repeating key experiments, possibly in vitro only (e.g., enrichment of LGR5, induction of WNT3 and beta catenin etc.) with one of these drugs, may be beneficial to broaden the conclusions of this manuscript to other relevant drugs that are used for CRC patients as well.

Minor issues

Text issues:

1. Lines 157-162 – the authors mention p21 as mediator of p53-induced apoptosis, and after performing experiments with p21 null isogenic cell-lines they conclude that p53 mediates its effects "independently of its apoptotic pathway". Please note that p21 is implicated in cell-cycle arrest rather than apoptosis (7,8) . Please correct the text accordingly.

2. Line 176-177 – the authors state: "virtual prediction identified p53 as one of the top putative transcription factors for WNT3 but not for R-spondin1". Do the authors mean an in-silico analysis? If so, it may be beneficial to include this analysis, either in the main figures or possibly in the supplementary material. In addition, it would be worthy to mention in this part, again, the findings of Wang et al.1, which previously identified WNT3 as a WT p53 target, although in a different system, as this may serve as a lead to examine WNT3 in the authors' system as well.

3. Lines 276-284 – please provide references for the statements regarding p53.

4. Lines 308-309 – there seems to be a duplication in the phrase "in the recurrence after 5-FU treatment". In general, it is advisable to proofread the manuscript before publication.

Figure issues:

5. Figure 2e – please indicate which p53 mutation (i.e. what is the amino acid substitution in the mutant p53 construct) was transfected to the HCT116 cells in the MUT condition. In addition, please indicate in the methods section the source of the plasmids which were used in this experiment, as well as how transfection was performed.

6. Figure 3i – please indicate which stainings were done (most probably similar to panel h) in the figure itself.

7. Figure legends 3g and 4h indicate that "immunoblots of extracts from LoVo and HCT116 cell extracts" and "ChIP-qPCR analysis ... in Lovo and HCT116 cells" while in both cases the panels seem to indicate results of only one cell-line.

8. Figure 4i+j – it seems that in the ICC images shown for APCKO/p53KO in panel 4i there is still some staining in beta catenin and not in WNT3, while in the quantification shown panel 4j it seems the other way around. Please make sure that there is no confusion in the quantifications, and if indeed the quantifications are correct, please provide more representative pictures which will correspond to the quantifications shown.

9. Figure 5e – the tumor shown for 5-FU+LGK974 before drug withdrawal looks larger than the tumor for the same treatment group after drug withdrawal, despite the fact that in panel 5d it is shown that the tumors grew to some extent after drug withdrawal. Please show more representative pictures.

10. Figure 5b-d,6f-i – for the ease of read, please indicate the cell-line/patient derived cell-line that was used for the experiments also in the figure itself and not only in the figure legend.

Relevant references

1. Wang, Q. et al. The p53 Family Coordinates Wnt and Nodal Inputs in Mesendodermal Differentiation of Embryonic Stem Cells. *Cell Stem Cell* 20, 70–86 (2017).

2. Kim, N. H. et al. p53 regulates nuclear GSK-3 levels through miR-34-mediated Axin2 suppression in colorectal cancer cells. *Cell Cycle* 12, 1578–1587 (2013).

3. Wang, J., Shou, J. & Chen, X. Dickkopf-1, an inhibitor of the Wnt signaling pathway, is induced by p53. *Oncogene* 19, 1843–1848 (2000).

4. Cha, Y. H. et al. miRNA-34 intrinsically links p53 tumor suppressor and Wnt signaling. *Cell Cycle* 11, 1273–1281 (2012).
5. Okayama, S. et al. P53 protein regulates Hsp90 atpase activity and thereby wnt signaling by modulating Aha1 expression. *J. Biol. Chem.* 289, 6513–6525 (2014).
6. Cagatay, T. & Ozturk, M. P53 Mutation As a Source of Aberrant B-Catenin Accumulation in Cancer Cells. *Oncogene* 21, 7971–7980 (2002).
7. Xiong, Y. et al. p21 is a universal inhibitor of cyclin kinases. *Nature* 366, 701–704 (1993).
8. Harper, J. W., Adami, G. R., Wei, N., Keyomarsi, K. & Elledge, S. J. The p21 Cdk-interacting protein Cip1 is a potent inhibitor of G1 cyclin-dependent kinases. *Cell* 75, 805–16 (1993).

Point-by-point response to the reviewers

We appreciate the reviewers for their constructive comments for improvement of the manuscript. The revised manuscript is substantially improved by performing new experiments in response to the reviewer's comments. We hope that our revised manuscript is now suitable for publication in *Nature Communications*.

Reviewers' comments:

Reviewer #1 (Remarks to the Author); expert in 5-FU resistance:

Using multiple in vitro and in vivo model system authors wanted to show that the presence of cancer stem cells in the bulk of cancer cells is the main reason for 5-FU resistance in chemotherapy. They also prove that WNT inhibitor along with 5FU increases the sensitivity of the treatment. They conclude that p53 is the critical mediator of 5-FU-induced CSC activation via the WNT/ β -catenin signaling pathway.

I have some concern of the MS

1) It will be better if authors make 5-FU resistance cell lines using different APC and P53 CRC cells and then carried out the experiment instead of 5FU treatment.

In this study, given that even in the patients who are sensitive to 5-FU, recurrence occurs due to enrichment of cancer stem cells (CSCs) and we demonstrated p53-dependent activation of WNT/ β -catenin signaling as the mechanism for the CSC enrichment by 5-FU in 5-FU-sensitive CRC patients. Our study is focusing on the identification of the CSC activation mechanism by 5-FU to improve the clinical outcome of 5-FU-sensitive CRC patients, rather than the identification of the 5-FU resistance mechanism.

However, as the reviewer mentioned, many other studies demonstrates that CSC are more resistant to 5-FU although the underlying mechanism for the resistance is not clearly identified. In line with those findings, upon repeated treatment of 5-FU, 5-FU-induced CSC activating mechanism may render the CRC cells more resistant to 5-FU treatment due to the CSC enrichment among the cancer cell population. Therefore, we thought that it would be more meaningful if we assess whether the 5-FU-induced CSC activation mechanism that we identified involves resistance of 5-FU.

As the reviewer suggested, we established 5-FU-resistant CRC cell lines with different APC and p53 mutations by culturing surviving clones after serial treatment of 5-FU¹ and addressed following comments. The establishment of 5-FU-resistant cells (5-FU-R) from HCT116 (APC wild-type, p53 wild-type), SW480 and HT-29 (APC mutant, p53 mutant) cell

lines was confirmed by cell growth after treatment of 5-FU (Fig. R1a). Interestingly, only *p53* wild-type HCT116-5-FU-R cells exhibited elevated β -catenin and WNT3 levels compared with its parental counterpart while 5-FU-R-SW480 and 5-FU-R-HT-29 cells which harbor mutant *p53* do not show changes in the β -catenin and WNT3 levels (Fig. R1b). These results show that the CSC activation mechanism we identified in 5-FU-sensitive *p53* wild-type CRC cells can also be applied in 5-FU resistant *p53* wild-type, but not *p53* mutant CRC cells.

Fig. R1 *p53* wild-type 5-FU-R-HCT116 cell line exhibits elevated β -catenin and WNT3 levels compared with its parental counterpart. a MTT analyses using 5-FU-resistant HCT116, SW480, and HT-29 cell lines and their parental counterparts after treatment of 5-FU (1.5 μ g/ml, 72 h). **b** Immunoblots of indicated proteins in 5-FU-resistant HCT116, SW480, and HT29 and their parental counterparts.

2) They need to show that enrichment of CSC happen due to 5 fu resistance, for that they need to make the CSC cells from 5-fu resistance cells and then characterized.

As the reviewer commented, we performed spheroid culture using the parental-and 5-FU-R-HCT116 cells to assess whether enrichment of CSC occurs in 5-FU-resistant cells. Consistent with the elevated β -catenin level, 5-FU-R-HCT116 cells exhibited increased cancer stemness as indicated by the increase of sphere forming ability (Fig. R2a).

3) For the mechanistic study the development of CSC model from 5-FU resistance cell will be better than transient treatment with 5FU.

As the reviewer suggested, we established CSC-like spheroid model from 5-FU-resistant HCT116 cells. Consistent with the CSC activation mechanism by 5-FU in 5-FU-responsive wild-type *p53* CRC, we found that the increments of β -catenin level and cancer stemness by 5-FU treatment in 5-FU-R-HCT116 cells were inhibited by treatment of the WNT inhibitor, LGK-974 (Fig. R2b). These results suggest that our mechanism can also be applied to the 5-FU-resistant wild-type *p53* CRC.

3) Why they only focus on WNT signaling, not others potential CSC signaling

Given that the CSC markers *LGR5*, *CD44* and *CD133* are direct target genes of the WNT/ β -catenin signaling pathway, the most crucial and fundamental signaling pathway for the CSC activation and maintenance in CRC is the WNT/ β -catenin pathway although other signaling pathways can also be involved in the regulation of CSC.

Throughout our study, we observed correlative increase in the expression of β -catenin and *LGR5* by treatment of 5-FU and almost complete inhibitory effect of WNT inhibitor when treated with 5-FU. Since NOTCH signaling is considered as the second most important signaling pathway for CSC regulation, we assessed the inhibitory effect of NOTCH inhibitor on CSC activation in 5-FU-resistant cells as well as CSC activation by transient treatment of 5-FU in 5-FU-sensitive cells to address the reviewer's query. We did not observe significant change of spheroid forming ability as well as β -catenin level and mRNA expression levels of the *LGR5*, *CD44*, *CD133*, and *CD166* CSC markers by treatment of NOTCH inhibitor, DAPT, in 5-FU-R-HCT116 cells (Fig. R2 a-c). Consistently, the activation of the WNT/ β -catenin pathway and CSCs by transient treatment of 5-FU in 5-FU-sensitive HCT116 cells were not inhibited by treatment of DAPT (Fig. R2 d, e). These results indicate that the WNT/ β -catenin pathway plays a major role in the 5-FU-induced CSC activation in both 5-FU-sensitive and -resistant *p53* wild-type CRC cells.

Fig. R2 WNT inhibitor, but not the NOTCH inhibitor, suppresses the cancer stemness of 5-FU-R-HCT116 cells.

a-c Parental and 5-FU-R-HCT116 cells were seeded at a density of 1×10^4 cells/mL in ultra-low attachment plates for spheroid culture for 5 days with or without treatment of LGK-974 (5 μ M) and DAPT (5 μ M). **a** Bright field images of spheroids derived from parental and 5-FU-R-HCT116 cells with indicated treatment (upper panel). Lower panel represents quantification results of their spheroid forming abilities as analyzed by using image J software. **b** Immunoblots of indicated proteins in spheroids derived from parental and 5-FU-R-HCT116 cells with indicated treatment. **c** Relative mRNA levels of *LGR5*, *CD44*, *CD133*, and *CD166* in parental and 5-FU-R-HCT 116 spheroids with indicated treatment. **d** Immunoblots of indicated proteins in HCT116 cells treated with or without 5-FU (1.5 μ g/ml, 48 h) alone or co-treated with DAPT (5 μ M, 48 h). **e** Relative mRNA levels of *LGR5*, *CD44*, *CD133*, and *CD166* in HCT 116 cells treated with or without 5-FU (1.5 μ g/ml, 48 h) alone or co-treated with DAPT (5 μ M, 48 h). Data are mean \pm s.d., two-sided Student's t-test, n.s. not significant, * $p < 0.05$, ** $p < 0.01$, *** $p < 0.001$.

4) Why they use 5Fu in combination with WNT inhibitor to increase the sensitivity of 5FU resistance. if the tumor already 5FU resistance then how with 5FU it will be sensitive.

Our current study demonstrates 5-FU-induced CSC activation in 5-FU-sensitive CRC as the potential mechanism for the recurrence after treatment of 5-FU rather than the mechanism for the 5-FU resistance. However, to address the reviewer's suggestion, we established 5-FU-resistant (5-FU-R) CRC cells and found out that the CSC activation

mechanism that we identified by transient treatment of 5-FU in 5-FU-sensitive *p53* wild-type CRC is involved in HCT116-5-FU-R cells. To test whether combined WNT inhibition with 5-FU can increase the sensitivity of 5-FU-R-HCT116 cells against the 5-FU treatment, we treated two different WNT inhibitors, LGK-974 and IWP-2, in combination with 5-FU. Despite their effective suppression of the 5-FU-induced WNT/ β -catenin pathway in 5-FU-R-HCT116 cells (Fig. R3a), both LGK-974 and IWP-2 reduced the cell viability of 5-FU-R-HCT116 cells by only 23.4 % and 27 %, respectively without the enhancement of 5-FU efficacy (Fig.R3b). Considering the effective inhibition of the activation of the WNT/ β -catenin pathway and CSC enrichment by co-treatment of WNT inhibitor with 5-FU in 5-FU-R-HCT116 cells (Fig. R2 a-c), 5-FU-resistance in this HCT116 cells was not solely due to the enrichment of CSCs and other driving factors might also be involved.

Fig. R3. WNT inhibitors partially enhance the inhibitory effect of 5-FU on the cell viability of *p53* wild-type 5-FU-resistant HCT116 cells.

a Immunoblots of indicated proteins in 5-FU-R-HCT116 cells treated with or without 5-FU (1.5 μ g/ml, 48 h) alone or co-treated with 5-FU (1.5 μ g/ml, 48 h) and LGK-974 (5 μ M, 48 h). **b** MTT analyses of 5-FU-R-HCT116 cells treated with or without 5-FU (1.5 μ g/ml, 72 h) alone or co-treated with 5-FU (1.5 μ g/ml, 72 h) and LGK-974 (5 μ M, 72 h). Data are mean \pm s.d., two-sided Student's t-test, n.s. not significant, ** $p < 0.01$, *** $p < 0.001$.

Reviewer #2 (Remarks to the Author); expert in Wnt signalling:

In this report, the authors investigated the effects of a chemo agent 5-FU on colorectal cancer stem cells (CSCs) and asserted that 5-FU stimulates production of Wnt3 in p53 WT colorectal cancer (CRC) cells, which in turn via β -catenin drives Lgr5 and other CSC stem cell marker expression and renders the tumor resistance to 5-FU, leading to CRC recurrence. While there are some intriguing findings revealed by the study, there are many unanswered questions that would mis-lead readers or obscure the real mechanisms.

1) It seems reasonably convincing that 5-FU drives Wnt3 expression in p53 WT cells. However, it becomes murky how or whether 5-FU can drive β -catenin increase or nature/mechanism of this increase. It has been shown by many that Wnt does not drive substantial β -catenin stabilization in APC-loss cells including CRC cells. Would the conclusion of this study be only applied to CRCs with functional APC? It seems that 5-FU increases total β -catenin in APC loss cells (LoVo). Are there alternative mechanisms for the increases in β -catenin levels (see #2)

As the reviewer commented, we also assumed that WNT ligand may not significantly affect the WNT/ β -catenin signaling pathway in *APC* loss CRC cells; however, WNT ligand induction by 5-FU further activated the WNT/ β -catenin pathway not only in human *APC* loss cell line, LoVo (Supplementary Fig. 5h), but also in *APC*-mutant driven murine intestinal tumor organoid and *APC*-loss human organoids (Figs. 2 a-d; 4h; 5a; 6a).

Many recent studies revealed that despite the presence of *APC* mutation in CRC, heterogeneous WNT pathway activity, the β -catenin paradox, exists and WNT3 is required for high activation of the WNT/ β -catenin signaling in *APC*-mutated CRC cells^{2,3}.

In this revision, to confirm the effect of WNT3 on the CRC cells with aberrant WNT/ β -catenin signaling attributed to the mutations of downstream components of WNT pathway, we treated recombinant human WNT3 to HCT116 and LoVo, the CRC cell lines. Treatment of WNT3 activated LRP6 as shown by the increased level of phosphorylated LRP6 and subsequently increased β -catenin level and significant increase in the active β -catenin levels confirmed the activation of the WNT/ β -catenin signaling pathway (Fig. R4a). Consistently, WNT3 treatment increased mRNA levels of CSC markers (Fig. R4b). These results indicate that increased WNT3 can further activate the WNT/ β -catenin pathway and enrich the CSCs in the CRC cells-

Fig. R4 Exogenous WNT3 activates the WNT/β-catenin pathway and enriches CSCs in HCT116 and LoVo, the CRC cell lines.

a Immunoblots of indicated proteins in HCT116 and LoVo cells with or without WNT3 treatment (50 ng/ml, 48 h). **b** Relative mRNA levels of *LGR5*, *CD44*, *CD133*, and *CD166* in HCT116 and LoVo CRC cell lines after WNT3 treatment (50 ng/ml, 48 h). Data are mean ± s.d., two-sided Student's t-test, *** $p < 0.001$.

2) The Wnt-β-catenin pathway regulates the stability of β-catenin. Total cellular β-catenin can be regulated not only by Wnt. Thus, cytosolic or nuclear β-catenin contents have to be examined if Wnt signaling is claimed. Along the same line, β-catenin mRNA and cadherin pathway need to be examined to determine if alternative mechanisms are involved.

As the reviewer suggested, we quantified cytosolic and nuclear β-catenin expression using immunocytochemistry in *p53* wild-type PDTOs, and additionally detected E-cadherin protein and mRNA levels of β-catenin in CRC cell lines. Strong increase in the cytosol and nuclear β-catenin by 5-FU indicates that 5-FU-induced activation of canonical WNT/β-catenin pathway in PDTOs with wild-type *p53*. No changes in the E-cadherin protein and β-catenin mRNA levels in the CRC cell lines confirm that alternative mechanisms are not involved in the activation of the WNT/β-catenin pathway by 5-FU (Fig. R5 a-c).

Fig. R5 5-FU activates canonical WNT/β-catenin signaling in CRC cell lines and PDTOs harboring wild-type *p53*.

a Immunoblots of indicated proteins in HCT116 and LoVo cells with or without 5-FU treatment (1.5 μg/ml, 48 h) alone or co-treatment with LGK974 (5 μM, 48h). **b** Relative mRNA levels of *β-catenin* in the HCT116 and LoVo cells with or without 5-FU (1.5 μg/ml, 48 h) alone or co-treatment with LGK974 (5 μM, 48 h). **c** Confocal images of immunofluorescence staining of E-cadherin (red) and β-catenin (green) in *p53* wildtype PDTOs followed by 5-FU treatment (1.5 μg/ml, 48 h) alone or co-treatment with LGK974 (5 μM, 48 h). (left panel). Quantification of the mean intensity of nuclear and cytosolic β-catenin were analyzed by Zen 3.1 software. Scale bars=50 μm. Data are mean ± s.d., two-sided Student's t-test, n.s. not significant, **p*<0.05, ****p* < 0.001.

3) What exactly the effects of 5-FU on Lgr5+ cells, converting non-Lgr5 into Lgr5+ cells or Lgr5+ cells are more resistant to 5-FU. Careful analysis of proliferation and apoptosis of Lgr5+ cells probably by flow cytometry, together with some kind lineage tracing means for Lgr5 conversion, is needed to address this question.

We appreciate the reviewer's fundamental comments. To determine whether 5-FU-mediated enrichment of Lgr5+ cells was resulted from the resistance of Lgr5+ cells to 5-FU or the conversion of Lgr5- cells into Lgr5+ cells, we sorted Lgr5+ or Lgr5- cells from the *Apc^{Min/+}/Lgr5^{EGFP}* tumor organoids by FACS and treated 5-FU. Similar with the results of others' recent studies^{4,5}, in the presence of 5-FU, Lgr5+ cells were enriched in the tumor

organoids derived from $Lgr5^-$ cells, and treatment of 5-FU to $Lgr5^+$ cells increased the expression of $Lgr5^+$ while effective induction of apoptosis of $Lgr5^+$ cells were observed (Fig. R6 a, b). Although we did not confirm these effects using lineage tracing models, our results suggest that the 5-FU-induced enrichment of $Lgr5^+$ cells occurs by dedifferentiation of $Lgr5^-$ cells and activation of the $Lgr5^+$ cells rather than by the resistance of $Lgr5^+$ cells to the 5-FU therapy.

Fig. R6 5-FU treatment enriches $Lgr5^+$ cells in tumor organoids derived from both $Lgr5^-$ and $Lgr5^+$ cells and induces apoptosis in $Lgr5^+$ cells sorted from the $Apc^{Min/+}/Lgr5^{EGFP}$ tumor organoids.

a, b $Lgr5^+$ cells or $Lgr5^-$ cells of tumor organoids derived from $Apc^{Min/+}/Lgr5^{EGFP}$ were sorted by FACS. **a** Immunocytochemical analyses of indicated proteins in tumor organoids derived from $Lgr5^-$ or $Lgr5^+$ cells with or without 5-FU treatment (1.5 $\mu\text{g}/\text{ml}$, 48 h). Scale bars represent 50 μm . **b** Immunoblots of indicated proteins in $Lgr5^+$ cells sorted from $Apc^{Min/+}/Lgr5^{EGFP}$ tumor organoids treated with or without 5-FU (1.5 $\mu\text{g}/\text{ml}$, 48 h).

4) Imaging-based quantifications are not reliable, unless ratio imaging is used. Western should be used in all cases.

We provide immunoblot analysis data for most of the cases throughout the manuscript except for the CSC markers, which we believe more accurately quantified by measurement of mRNA levels or GFP detection by immunohistochemistry/

immunocytochemistry analyses. As the reviewer recommended, most of the queries raised by the reviewers have been addressed by performing additional immunoblot analyses.

5) The increase in mRNA of Wnt3 in HCT116 does not seem to be pronounced. Can the effect of 5-FU be recapitulated by adding exogenous Wnt? Comprehensive analysis of the gene expression of 5-FU-treated cells may be needed to gain a more complete picture. Wnt3 may be only a small part of the truth.

To investigate whether exogenous WNT can recapitulate the effect of 5-FU, we assessed the activation of the WNT/ β -catenin pathway and CSC markers after treatment of WNT3 in HCT116 and LoVo CRC cells. Treatment of exogenous WNT3 significantly activated the WNT/ β -catenin pathway in CRC cells, and increased mRNA levels of CSC markers (Fig. R7 a, b). These results show that WNT3 can recapitulate the effects of 5-FU. Moreover, we confirmed the secretion of WNT3 by 5-FU treatment in HCT116 and LoVo cells (Fig. R7c). We agree that comprehensive analysis of the gene expression after treatment of 5-FU can provide a more complete understanding of 5-FU response in CRC; however, recapitulation of 5-FU effects by exogenous WNT3 and complete suppression of 5-FU-induced activation of the WNT/ β -catenin signaling pathway and CSC by co-treatment of WNT inhibitor with 5-FU indicate that, at least for the WNT/ β -catenin pathway activation and CSC enrichment by 5-FU, WNT3 induction plays a major role.

Fig. R7 Exogenous WNT3 activates the WNT/β-catenin pathway and CSCs in HCT116 and LoVo, the CRC cell lines.

a Immunoblots of indicated proteins in HCT116 and LoVo cells with or without WNT3 treatment (50 ng/ml, 48 h). **b** Relative mRNA levels of *LGR5*, *CD44*, *CD133*, and *CD166* in HCT116 and LoVo cell lines after WNT3 treatment (50 ng/ml, 48 h). **c** Measurement of WNT3 secretion by ELISA analyses using 12 h cultured medium of HCT116 and LoVo cells after treatment with 5-FU (1.5 μg/ml, 48 h) alone or co-treated with 5-FU (1.5 μg/ml, 48 h) and LGK-974 (5 μM, 48 h). Data are mean ± s.d., two-sided Student's t-test, ***p < 0.001.

6) Can the LGK-974 effect be reproduced in organoids derived from single APC-null stem cells? Similarly, can the data in Figure 5 using HCT116 be reproduced with APC-loss cells? Additional means to inhibit Wnt signaling is needed as LGK may have off-target effects.

As the reviewer commented, we confirmed the effect of LGK-974 on the inhibition of 5-FU-induced activation of the WNT/β-catenin pathway in human organoids derived from single APC-null stem cells. Treatment of 5-FU significantly increased β-catenin, and combined treatment of LGK-974 inhibited the activation of the WNT/β-catenin pathway by 5-FU (Fig. R8).

Fig. R8 LGK-974 effectively decreases 5-FU-induced activation of β -catenin in organoids derived from single *APC*-null stem cells. Single *APC*-null stem cells (*APC*^{KO} human intestinal stem cells) were grown using 3D organoid culture medium with or without 5-FU treatment (1.5 μ g/ml, 48 h) alone or co-treatment with LGK-974 (5 μ M, 48 h). Confocal images of immunofluorescence staining of β -catenin (green) using organoids derived from the *APC*-null stem cells. Scale bars=50 μ m.

To address the reviewer's concern that LGK-974 may have off-target effects, we assessed the effect of another WNT inhibitor, IWP-2. Consistent with LGK-974, treatment of IWP-2 inhibited 5-FU-induced activation of the WNT/ β -catenin pathway as shown by the total and active β -catenin levels (Fig. R9a). Treatment of IWP-2 blocked the transcriptional induction of LGR5, CD44, CD133 and CD166 CSC markers by 5-FU, showing that the 5-FU-mediated activation of CSCs occurs through WNT/ β -catenin pathway activation via WNT secretion (Fig. R9b).

Fig. R9 5-FU increased the stemness of CRC cells via activation of WNT/ β -catenin pathway.

a, b HCT116 and LoVo cells were treated with or without 5-FU (1.5 μ g/ml, 48 h) alone or co-treated with IWP-2 (5 μ M, 48 h). Immunoblots of indicated proteins (**a**) and relative mRNA levels of *LGR5*, *CD44*, *CD133*, and *CD166* (**b**) in HCT116 and LoVo cells after the indicated treatments. Data are mean \pm s.d., two-sided Student's t-test, * $p < 0.05$, ** $p < 0.01$, *** $p < 0.001$.

7) Why does β -catenin staining show punctate patterns in Fig. 6e rather than uniform as other staining in the study? Does it mean that cells in the organoids are heterogenous? The changes in β -catenin levels in fig 6d are less pronounced. It is hard to believe that the combo effects shown in Fig. 6h,I are exclusively due to these small differences in β -catenin levels, especially we do not even know what kind of β -catenin it is. The real mechanism could be more complex than what is asserted here.

We agree with the reviewer's comments. To improve the punctuated β -catenin expressions in Fig. 6e, we immuno-stained β -catenin using the whole patient-derived tumor organoids (PDTOs). As a result, we confirmed significant increase of nucleus and cytosolic β -catenin expression by 5-FU treatment and effective abolishment of the 5-FU-induced increment by co-treatment of WNT inhibitor LGK-974 with 5-FU (Fig. R10a). However,

PDTOs, which represent the characteristics of the CRC patients, consist of heterogenous population of cancer cells. Therefore, the heterogenous staining pattern shown in this study would be reasonable.

Fig. R10 LGK-974 inhibits 5-FU-induced activation of the WNT/ β -catenin signaling in *p53* wild-type patient-derived tumor organoids. PDTO#2 and #3 were grown using 3D organoid culture medium treated with 5-FU (1.5 μ g/ml, 48 h) alone or co-treated with 5-FU (1.5 μ g/ml, 48 h) and LGK-974 (5 μ M, 48 h) (left panel). Confocal images of immunofluorescence staining of β -catenin (green) in PDTOs with indicated treatments. Scale bars=50 μ m.

Reviewer #3 (Remarks to the Author); expert in CRC organoids and cancer stem cells:

This is an interesting paper in which Cho and colleagues suggest a potential mechanism of acquired 5FU resistance via p53 mediated Wnt activation. Overall the data is of a good quality and clearly laid out and the results point to a potential way to overcome 5-FU resistance in some colorectal cancer patients. I have a number of comments on the study and some experiments I think are important for supporting the conclusions of the work.

Major points

1) The overall conclusions of the study seem counter intuitive. In the introduction the authors outline some known functions of P53 in response to 5FU. All of these are tumour suppressive but this study suggests the opposite, that P53 is an important mediator of disease relapse. Due to the apparent contradiction, more supporting evidence from human patient cohorts is needed. For example, are patients with wild type P53 more likely to relapse? Do they have

shorter survival times than P53 mutant patients? What does Wnt signaling look like in these patients?

As the reviewer mentioned, the tumor suppressive roles of p53 in response to 5-FU have been well characterized and as we stated in our manuscript, CRC patients with deficiency or mutation in *p53* show decreased responsiveness to the 5-FU therapy, leading to poor clinical outcome. However, in this study, we identified p53-mediated activation of the WNT/ β -catenin signaling pathway as the underlying mechanism for the recurrence after treatment of 5-FU in 5-FU-responsive *p53* wild-type CRC rather than the mechanism associated with overall disease free survival of 5-FU therapy received CRC patients. Although we also believe that more supporting evidence that links recurrence and WNT signaling from 5-FU-treated *p53* wild-type patient cohorts would consolidate our findings, comparison of the relapse frequency between *p53* wild-type and *p53*-mutant patients would not provide pathophysiological/clinical evidence for our mechanism. Our study reveals the two faces of p53 involving both anti-cancer effects and CSC-inducing effects under treatment of 5-FU in *p53* wild-type 5-FU-responsive CRC, but we do not assume the status of *p53* itself as a predictive marker for CRC relapse due to the complexities of genetic backgrounds and therapy treatments in patients.

In addition, unfortunately we were unable to achieve the differential status of WNT signaling in *p53* mutant patients; however, in an effort to address the reviewer's query, we assessed the basal levels of β -catenin and WNT3 in *p53* wild-type (+/+), knockout (-/-) and mutant (R248W/-) isogenic CRC cell lines. There was no significant difference in the basal β -catenin levels among the three cell lines while the basal WNT3 levels were significantly lower in *p53* knockout and mutant cells compared with *p53* wild-type cells, indicating that in unstimulated cells, other factors might be involved in the regulation of the WNT/ β -catenin pathway by p53. Consistent with our study, treatment of 5-FU induced WNT3 expression and activation of the WNT/ β -catenin pathway in *p53* wild-type cells but not in *p53* knockout and mutant cells, showing that *p53*-mediated WNT3 induction majorly regulates the activation of the WNT/ β -catenin pathway upon stimulation of wild-type *p53* by 5-FU (Figure R11). Together, these results suggest that the mechanism identified in this study involving the CSC activation will be helpful to understand the CSC-mediated recurrence after 5-FU therapy in 5-FU-responsive CRC patients harboring wild-type *p53* but not the overall disease free survival of 5-FU therapy-received CRC patients.

Fig. R11 p53 mediates 5-FU-induced activation of the WNT/β-catenin signaling pathway. Immunoblots of indicated proteins in $p53^{+/+}$, $p53^{-/-}$, and $p53^{R248W/-}$ isogenic HCT116 cell lines treated with or without 5-FU (1.5 μg/ml, 48 h).

2) The authors demonstrate that following 5FU treatment, organoids exhibit increased stem cell marker expression. What about stem cell function? Clonogenicity assays should be carried out to assess if these are functional stem cells.

As the reviewer mentioned, consistent with the increased stem cell marker expression following 5-FU treatment, the stem cell function was also increased as shown by the repopulation of the tumor organoids after withdrawal of 5-FU (Fig. 6h). To further address the reviewer's comment, we also performed clonogenic assay with the $p53$ wild-type CRC cell line to confirm the 5-FU-induced CSC activation mechanism. Consistent with our mechanism, treatment of LGK-974, which inhibits the 5-FU-induced activation of the WNT/β-catenin signaling followed by increase in CSC markers, effectively inhibited the colony forming ability after withdrawal of 5-FU (Fig. R12 a, b), indicating that WNT inhibition effectively suppresses the reconstitution ability of the surviving clones after 5-FU treatment.

Fig. R12 WNT inhibitor decreases 5-FU-mediated clonogenic ability. HCT116 cells were treated with 5-FU (1.5 $\mu\text{g}/\text{ml}$, 48 h) alone or combined with LGK-974 (5 μM , 48 h) 3 days after seeding. Colony forming ability were analyzed 10 days after discontinuation of indicated treatment. **a** Gross images of colony formation assay. **b** Relative quantification of colony forming ability. Data are mean \pm s.d., two-sided Student's t-test, * $p < 0.05$.

3) What does 5FU do to the organoids and why does Wnt signalling protect Lgr5⁺ stem cells? The authors should investigate apoptosis and proliferation in the 5FU treated organoids, in particular, are Lgr5⁺ stem cells more resistant to the effects?

To address the reviewer's question, we sorted Lgr5⁺ stem cells from *Apc^{Min/+}/Lgr5^{EGFP}* tumor organoids treated with or without 5-FU and investigated effects of 5-FU on its apoptosis by Western blot analyses of the sorted cells. Treatment of 5-FU effectively induced apoptosis of Lgr5⁺ cells sorted from murine CRC tumor organoids treated with 5-FU (Fig. R13a). Moreover, treatment of 5-FU enriched GFP-Lgr5⁺ cells from the sorted Lgr5⁻ organoid, indicating that 5-FU induces dedifferentiation of non-cancer stem cells while it further increased the expression of GFP-Lgr5 in Lgr5⁺ stem cells (Fig. R13b). Together, these results suggest that Lgr5⁺ cells are responsive to 5-FU treatment; however, WNT signaling

activation by 5-FU induces enrichment of Lgr5⁺ CSC cells in the tumor organoids.

Fig. R13 5-FU treatment induces apoptosis in Lgr5⁺ cells, and enriches GFP-Lgr5⁺ cells in tumor organoids derived from both Lgr5⁻ and Lgr5⁺ cells which were sorted from the *Apc*^{Min/+}/*Lgr5*^{EGFP} tumor organoids. Lgr5⁺ cells or Lgr5⁻ cells of tumor organoids derived from intestinal tumors of *Apc*^{Min/+}/*Lgr5*^{EGFP} were sorted by FACS. **a** Immunoblots of indicated proteins in Lgr5⁺ cells with or without 5-FU treatment (1.5 μ g/ml, 48 h). **b** Immunocytochemical analyses of indicated proteins in tumor organoids derived from Lgr5⁻ or Lgr5⁺ cells with or without 5-FU treatment (1.5 μ g/ml, 48 h). Scale bars represent 50 μ m.

4) Related to this, is Wnt3 sufficient to drive these phenotypes? Does Wnt3 of organoids / cell lines lead to the same Bcat stabilisation and stem cell marker increases? And does this impact on 5FU treatment response?

As the reviewer commented, we investigated whether WNT3 is sufficient to drive β -catenin stabilization and CSC marker increases using two CRC cell lines with wild-type *p53*. Treatment of WNT3 significantly increased the levels of both total and active β -catenin followed by increment of the CSC markers (Fig. R14 a, b). The WNT/ β -catenin signaling activation by as shown by increment of both β -catenin and active β -catenin levels by 5-FU treatment was mostly abolished, indicating that WNT induction is the major influencing factor for the 5-FU induced β -catenin increase (Fig. R14c).

Fig. R14 Exogenous WNT3 activates the WNT/β-catenin pathway and CSCs, and WNT inhibitor abrogates 5-FU-induced activation of the WNT/β-catenin pathway. a Immunoblots of indicated proteins in HCT116 and LoVo cells with or without WNT3 treatment (50 ng/ml, 48 h). **b** Relative mRNA levels of *LGR5*, *CD44*, *CD133*, and *CD166* in HCT116 and LoVo CRC cell lines after WNT3 treatment (50 ng/ml, 48 h). **c** Immunoblots of indicated proteins in HCT116 and LoVo cells treated with 5-FU (1.5 μg/ml, 48 h) alone or co-treated with 5-FU (1.5 μg/ml, 48 h) and LGK-974 (5 μM, 48 h). Data are mean ± s.d., two-sided Student's t-test, ***p < 0.001.

Minor points

- 1) What is the dataset used for Fig S1? Is P53 status known and does this impact on survival?
The source of dataset used for Fig. S1 is indicated in the Material and method section of the revised manuscript. Unfortunately, the *p53* status of patients was not provided, and we were unable to analyze its correlation with the survival of patients.
- 2) The IHC images for Fig 2 are poor quality and better resolution images are needed.
As reviewer suggested, we have improved the quality of images with better resolution, and replaced with the new ones in the revised manuscript.
- 3) The cell line data in Fig 3a is not clear to me. It looks like the P53 mut cell lines have higher basal B-cat than the p53 wt. Does this suggest a p53 independent mechanism of increasing Wnt signaling in these cells lines?

The β -catenin levels of *p53* wild-type and mutant cells in Fig. 3a cannot be compared because they were ran on different gels. As we have addressed in the response to major point 1, there was no significant difference in the basal β -catenin level among the *p53* wild-type, knockout and mutant isogenic cell lines (Fig. R11). According to this result, we assume that, in addition to the WNT3 regulation by *p53*, other mechanism mediated by *p53* may be involved in the negative regulation of the basal β -catenin level, as shown by previous studies ^{6, 7, 8, 9, 10}. Therefore, *p53* may play roles as both negative and positive regulators of the WNT/ β -catenin pathway in the intact status; however, upon 5-FU treatment, induction of WNT3 by activation of wild-type *p53* majorly affects the WNT/ β -catenin pathway.

Reviewer #4 (Remarks to the Author); expert in *p53* and cancer stem cells:

In the current study, Cho and colleagues aim to discover what governs tumor recurrence after 5-FU treatment in colorectal cancer (CRC) patients. They hypothesize that tumors recur after 5-FU treatment due to expansion of cancer stem cells (CSCs) population, and further aim to elucidate the molecular mechanisms that govern this expansion and tumor recurrence. They begin by demonstrating that indeed 5-FU treatment, while inhibiting tumor cell growth, enriches the CSC population in CRC organoids and murine tumors. They further demonstrate that this enrichment, which is caused by 5-FU treatment, is associated with increased beta-catenin expression, which is a hallmark of WNT pathway activation. Next, they show that the activation of beta catenin after 5-FU treatment is caused by *p53* activation. They next demonstrate that *p53* upregulates WNT3, which is associated with beta catenin upregulation. In addition, the authors demonstrate that the activation of the WNT pathway after 5-FU treatment can be inhibited by a WNT secretion inhibitor, which also inhibits the recurrence of tumors in vivo via inhibiting the activation of CSCs. Finally, the authors demonstrate the aforementioned findings also in patient-derived cell-lines and organoids, thus demonstrating potential clinical relevance.

These findings are indeed very intriguing, and demonstrate how WT *p53* activation, at least in colon cancer, can have negative implications on eventual tumor outcome, and suggest a potential therapeutic strategy to overcome them. This manuscript may be indeed of high interest to the scientific and medical community, provided that the following issues are addressed:

1. Figure 3 – In panel A, it is worthy to notice that although beta catenin is indeed

upregulated (compared to the basal level) after 5-FU treatment only in WT p53 expressing cells, it is noticeable that in mutant p53 expressing cells, beta catenin is already activated in basal level, although indeed there is no further activation after 5-FU treatment. As for comparison between the effects of different p53 status on an isogenic cellular background of HCT116, there seems to be a discrepancy between the results shown in panel C and panel E. While panel C shows that in HCT116 p53^{-/-} there is upregulation of beta catenin already in basal level, without 5-FU treatment, it does not seem so in the experiment depicted in panel E. This aspect, although not pertaining to 5-FU response directly, is important in terms of p53 regulation of the WNT pathway, which is further important to the activation of CSCs which is a major part of this work. The authors do not address this point neither in the results section nor in the discussion section. Indeed, WNT3, which activates the WNT pathways in stem cells in a non-cell autonomous manner, was shown to be a WT p53 target both by the authors and by a previous publication that the authors reference in their manuscript (Wang et al (1)). However, several other publications show that WT p53 is a negative regulator of the WNT pathway, in a cell autonomous manner and via different mechanisms (2–5), and that mutant p53 may even upregulate beta-catenin (6), which is in line with the aforementioned results shown in panels A and C. Therefore, I think several issues are needed to be addressed in that regard:

A. In panel 3C - were the HCT116^{-/-} ran on the same gel as HCT116^{+/+}? If not, please repeat the experiments and run on the same gel so beta catenin levels could be compared. Of note, it may be of value, if possible and obtainable, to include HCT116 p53^{R248W}^{-/-} mutant p53-expressing sub-line, which was also generated by prof. Bert Vogelstein's laboratory, in this experiment.

B. How do the authors explain the discrepancy between the experiments in panel C and E? If one of the experiments seems to be technically incorrect, please exclude it from the manuscript.

C. Please address the aforementioned known relations between the WNT signaling and p53 in the results and/or in the discussion section/s, and how do you interpret your results in light of this aforementioned literature.

Response to 1A-1C:

As the reviewer stated, several studies have shown the role of p53 in as a negative regulation of the WNT signaling pathway while recent studies by Wang et al.¹¹ has suggested

WNT3 as a target gene of p53.

Here, since our study investigates the activation of CSC by 5-FU treatment, we have focused on the effect of p53 activation on the WNT/ β -catenin signaling pathway rather than the effect by un-stimulated basal p53.

However, as the reviewer commented, the basal β -catenin levels of HCT116 *p53*^{-/-} and *+/+* cells in Fig. 3c (original unrevised manuscript), which were run on different gels, may confuse the effect of p53 activation on the β -catenin level. Therefore, following the reviewer's suggestion, we obtained HCT116 *p53R248W*^{-/-}-mutant *p53*-expressing sub-line from prof. Bert Vogelstein's laboratory and ran *p53*^{+/+}, *p53*^{-/-}, and *p53R248W*^{-/-} HCT116 isogenic cell lines on the same gel to clarify the differential basal β -catenin expression levels among the three isogenic cell lines (Fig. R15a).

Consistent with our previous results, treatment of 5-FU increased β -catenin and WNT3 levels in *p53*^{+/+} cells but not in *p53*^{-/-} and *p53R248W*^{-/-} cells, showing that 5-FU activates the WNT/ β -catenin pathway via p53-mediated WNT3 induction. Interestingly, there was no significant difference in the basal β -catenin level among the three isogenic cell lines while the basal WNT3 levels were significantly lower in *p53*^{-/-} and *p53R248W*^{-/-} cells compared with that in *p53*^{+/+} cells. We assume that other mechanism mediated by p53 in addition to the WNT3 regulation by p53 may be involved in the negative regulation of the basal β -catenin level, as shown by previous studies^{6,7,8,9,10}. Therefore, p53 may play roles as both negative and positive regulator of the WNT/ β -catenin pathway in the intact status; however, upon 5-FU treatment, induction of WNT3 by activation of wild-type p53 majorly affects the WNT/ β -catenin pathway.

To eliminate any confusion, we replaced Fig. 3c by Fig. R15a, and Figure 4c by Fig. R15b. We also addressed the aforementioned known relations between the WNT signaling and p53 and our interpretation of the results in light of this aforementioned literature in the result and discussion sections.

a**b**
Fig. R15 p53 mediates 5-FU-induced activation of the WNT/β-catenin signaling pathway.
a Immunoblots of indicated proteins in $p53^{+/+}$, $p53^{-/-}$, and $p53^{R248W/-}$ isogenic HCT116 cell lines treated with or without 5-FU (1.5 μg/ml, 48 h). Relative mRNA levels of *WNT3* in $p53^{+/+}$, $p53^{-/-}$, and $p53^{R248W/-}$ isogenic HCT116 cell lines after 5-FU treatment (1.5 μg/ml, 48 h). Data are mean ± s.d., two-sided Student's t-test, n.s., not significant. *** $p < 0.001$.

1D. According to the current authors' interpretation, only patients that harbor WT p53 may benefit from a combined 5-FU and WNT inhibitor treatment. However, as mentioned above, mutant p53 expressing cells may also exhibit increased WNT activation, which may lead to increased CSC population (although this was not addressed in the current manuscript), independently of 5-FU treatment. Therefore, do the authors indeed expect that the combined treatment will be applicable only for patients harboring tumor with intact WT p53? Please address this issue in the discussion section as well.

As we have addressed in the response 1A-C, our manuscript focuses on the activation of the WNT/β-catenin pathway and CSC enrichment by 5-FU-induced activation of wild-type p53. In cells harboring null or mutant *p53*, 5-FU treatment does not induce WNT ligand and does not activate WNT/β-catenin pathway. Moreover, the basal levels of β-catenin and CSC markers were not increased by *p53* mutation as shown by HCT116 isogenic cell lines. Therefore, if activation of WNT/β-catenin signaling pathway and CSC enrichment were observed in *p53* mutant cells, other factors might be involved the regulation. Hence, we

expect the combined treatment of WNT ligand inhibitor and 5-FU would only be effective in patients harboring wild-type *p53*. As the reviewer commented, we have addressed this issue in the discussion section as well.

2. It seems to me that a more accurate way to quantify WNT pathway activation, which leads to expansion of CSCs (e.g. in figure 2B+D, figure 3b, 3h-I etc.), would be by assessing percentage of (nuclear?) beta catenin positive cells (similar to what the authors did for LGR5 in panel 1d), rather than the mean intensity of the protein levels, which could reflect a stronger activation of beta catenin in the same cells.

As the reviewer commented, we quantified the WNT pathway activation by assessing the percentage of β -catenin-positive cells in murine and patient-derived tumor organoids (Fig. R16) and included the data in the revised manuscript (Fig.2b; Fig.6e).

Fig. R16 5-FU increases the percentage of β -catenin-positive cells in both murine and patient-derived tumor organoids. a, b Quantification of the β -catenin⁺ cells in the murine (a)- and CRC patient (b)-derived tumor organoids were measured by Zen software. a *Apc*^{Min/+}/*Lgr5*^{EGFP} intestinal tumor organoids were treated with or without 5-FU (1.5 μ g/ml,

48 h). **b** *p53* wild-type PDOs, PDO#2 (left panel) and PDO#3 (right panel), were treated with 5-FU (1.5 $\mu\text{g/ml}$, 48 h) alone or co-treated with 5-FU (1.5 $\mu\text{g/ml}$, 48 h) and LGK-974 (5 μM , 48 h). Data are mean \pm s.d., two-sided Student's t-test, ** $p < 0.01$. *** $p < 0.001$.

3. Figure 4i - It would be beneficial to assess secreted WNT3 protein levels (e.g. by ELISA) and not only WNT3 levels by IHC, which could represent WNT3 which is accumulated the cells that was not secreted. This is also relevant for the experiment described in figure 5A-C, in order to assess whether LGK-974 affected the levels of secreted WNT3, therefore demonstrating that the associated phenotypes are due to depletion of secreted WNT3.

As the reviewer commented, we also think that it will be clearer if we could assess the secreted WNT3 protein levels; however, we had hard time assessing the secretion of WNT3 using organoids due to the matrigel matrix used in the organoid culture.

In an effort to address the reviewer's point, we assessed the effect of *p53* on WNT3 secretion using isogenic *p53* $+/+$ and $-/-$ CRC cell lines instead of *p53* $+/+$ and *p53* $-/-$ human organoids. Consistent with our IHC results in Fig. 4h, treatment of 5-FU significantly increased WNT3 secretion in *p53* $+/+$ cells, but not in *p53* $-/-$ cell (Fig. R17a), suggesting that *p53* is major factor for regulating the expression of WNT3 in the CRC cells treated with 5-FU. Also, the inhibitory effect of LGK-974 on the 5-FU-induced WNT3 secretion was confirmed as shown by Fig. R17b. These results confirm that 5-FU activates the WNT/ β -catenin pathway and CSC enrichment by *p53*-mediated WNT3 induction followed by its secretion. We have added these results in the revised manuscript (Fig.5b; Supplementary Fig.5g).

Fig. R17 5-FU induces WNT3 secretion in a p53-dependent manner. **a** ELISA analysis of WNT3 secretion in the *p53* wild-type and *p53* knockout isogenic HCT116 cells using 12 h cultured medium with or without 5-FU treatment (1.5 $\mu\text{g/ml}$, 48 h). **b**. ELISA analyses of WNT3 secretion in the HCT116 and LoVo cells using 12 h cultured medium treated with 5-FU (1.5 $\mu\text{g/ml}$, 48 h) alone or co-treated with 5-FU (1.5 $\mu\text{g/ml}$, 48 h) and LGK-974 (5 μM , 48 h). Data are mean \pm s.d., two-sided Student's t-test, n.s., not significant. *** $p < 0.001$.

4. For the in-vivo experiment described in figure 5d-h – an assessment of p53 activation is missing to demonstrate the link between p53 and the induction of WNT signaling in-vivo. Please assess p53 activation by WB and/or IHC as depicted in panels 5f and 5h, respectively.

As the reviewer suggested, we detected p53 by western blot with improved quality of β -catenin and β -actin bands (Fig. R18a), and added the results in Fig. 5g of the revised manuscript.

Fig. R18 5-FU-induces activation of the WNT/ β -catenin signaling pathway via p53-mediated WNT induction in HCT116 xenografted tumor. HCT116 cells were xenografted in mice and tumors were extracted and analyzed after treatment with 5-FU alone or co-treatment with 5-FU (25 mg/kg) and LGK-974 (5 mg/kg) daily for 21 days. n=5 mice per group. Immunoblots of indicated proteins in isolated tumors after indicated treatment.

5. Figure 6 – please include at least one patient-derived cell-line and organoid with mutated p53 as a negative control for p53, WNT3 and LGR5 induction after 5-FU, at least for the in-vitro assays.

As the reviewer suggested, we investigated the effect of 5-FU on WNT3 and LGR5 induction in patient-derived cell line (PDC#2) harboring *p53* mutation. In *p53* mutant-PDC, treatment of 5-FU did not induce WNT3 expression and no changes in β -catenin (Fig. R19a). In addition, mRNA levels of *WNT3* and *LGR5* did not significantly change by treatment of 5-FU in *p53*-mutant PDC (Fig. R19b). These CRC patient results confirm clinical relevance of the p53-mediated WNT/ β -catenin signaling and CSC activation by 5-FU. We have added these results in the supplementary Figure (Fig. S6a, b) of the revised manuscript.

Fig. R19 5-FU-induced activation of the WNT/ β -catenin signaling pathway and subsequent CSC enrichment did not occur in the *p53* mutant CRC patient cell a Immunoblots of indicated proteins in #2 PDC harboring mutant *p53* after treatment of 5-FU (1.5 μ g/ml, 48 h). **b.** Relative mRNA levels of *WNT3* and *LGR5* in #2 PDC with or without 5-FU treatment (1.5 μ g/ml, 48 h). Data are mean \pm s.d., two-sided Student's t-test, n.s. not significant.

6. Based on the authors' results, especially those that demonstrate activation of beta catenin and CSC expansion after RITA treatment, it could be speculated that any compound or drug that activates *p53*, such as irinotecan or oxaliplatin, which are also used in CRC, could result in the same CSC enrichment and tumor recurrence. Do the authors expect this to be true? Perhaps repeating key experiments, possibly in vitro only (e.g., enrichment of *LGR5*, induction of *WNT3* and beta catenin etc.) with one of these drugs, may be beneficial to broaden the conclusions of this manuscript to other relevant drugs that are used for CRC patients as well.

As the reviewer commented, we also thought that it would be beneficial to broaden our results to other *p53* activating chemotherapies, oxaliplatin and irinotecan, that are widely used for CRC patients^{12, 13, 14, 15}. Therefore, we investigated the effect of oxaliplatin and irinotecan on the WNT/ β -catenin signaling and CSC markers using two *p53* wild-type CRC cell lines, HCT116 and LoVo. In both cell lines, treatment of oxaliplatin and irinotecan

significantly increased levels of β -catenin and WNT3, and mRNA levels of the CSC markers (Fig. R20a, b). Moreover, the increments of β -catenin and CSC markers by oxaliplatin or irinotecan were inhibited by treatment of LGK-974 (Fig. R21a, b), indicating that the mechanism that we identified using 5-FU can also be applied to oxaliplatin and irinotecan, and combined treatment of WNT inhibitor with these therapies may be therapeutically beneficial. We have added these results in the supplementary Figure (Fig. S7) of the revised manuscript.

Fig. R20 Oxaliplatin and irinotecan activate the WNT/ β -catenin pathway and increase stemness of CRC. **a-c** HCT116 and LoVo cells were treated with 5-FU (1.5 μ g/ml, 48 h), oxaliplatin (2.5 μ g/ml, 48 h), and irinotecan (10 μ M, 48 h). **a** Immunoblots of indicated proteins in HCT116 and LoVo cells after indicated treatment. **b-c** Relative mRNA levels of *WNT3* (**b**) and *LGR5*, *CD44*, *CD133*, and *CD166* CSC markers (**c**) in HCT116 and LoVo after indicated treatment. Data are mean \pm s.d., two-sided Student's t-test, ***p < 0.001.

Fig. R21 Activation of cancer stem cell by oxaliplatin and irinotecan was inhibited by treatment of WNT inhibitor. a, c Immunoblots of indicated proteins in HCT116 and LoVo cells treated with or without oxaliplatin (2.5 $\mu\text{g}/\text{ml}$, 48 h) alone or co-treated with oxaliplatin (2.5 $\mu\text{g}/\text{ml}$, 48 h) and LGK-974 (5 μM , 48 h) (a) and irinotecan (10 μM , 48 h) alone or co-treated with irinotecan and LGK-974 (5 μM , 48 h) (c). **b, d** Relative mRNA levels of *LGR5*, *CD44*, *CD133*, and *CD166* in HCT116 and LoVo cells after treatment with or without oxaliplatin (2.5 $\mu\text{g}/\text{ml}$, 48 h) alone or co-treatment with oxaliplatin (2.5 $\mu\text{g}/\text{ml}$, 48 h) and LGK-974 (5 μM , 48 h) (b) and irinotecan (10 μM , 48 h) alone or co-treated with irinotecan and LGK-974 (5 μM , 48 h) (d). Data are mean \pm s.d., two-sided Student's t-test, * $p < 0.05$, ** $p < 0.01$, *** $p < 0.001$.

Minor issues

Text issues:

1. Lines 157-162 – the authors mention p21 as mediator of p53-induced apoptosis, and after performing experiments with p21 null isogenic cell-lines they conclude that p53 mediates its effects “independently of its apoptotic pathway”. Please note that p21 is implicated in cell-cycle arrest rather than apoptosis (7,8) . Please correct the text accordingly.

As the reviewer suggested, we corrected the text accordingly in the revised manuscript.

2. Line 176-177 – the authors state: “virtual prediction identified p53 as one of the top putative transcription factors for WNT3 but not for R-spondin1”. Do the authors mean an in-silico analysis? If so, it may be beneficial to include this analysis, either in the main figures or possibly in the supplementary material. In addition, it would be worthy to mention in this part, again, the findings of Wang et al.1, which previously identified WNT3 as a WT p53 target, although in a different system, as this may serve as a lead to examine WNT3 in the authors’

system as well.

The prediction was not performed by us but was identified by QIAGEN as provided in the GeneCards database. As the reviewer commented, we mentioned the findings of Wang et al in the beginning of the Figure 4 result section of the revised manuscript.

3. Lines 276-284 – please provide references for the statements regarding p53.

We provided references for the statements regarding p53 in discussion section of the revised manuscript.

4. Lines 308-309 – there seems to be a duplication in the phrase “in the recurrence after 5-FU treatment”. In general, it is advisable to proofread the manuscript before publication.

As the reviewer recommended, we carefully proofread the manuscript and removed the duplicated phrase.

Figure issues:

5. Figure 2e – please indicate which p53 mutation (i.e. what is the amino acid substitution in the mutant p53 construct) was transfected to the HCT116 cells in the MUT condition. In addition, please indicate in the methods section the source of the plasmids which were used in this experiment, as well as how transfection was performed.

To address the major issues raised by the reviewers, we obtained p53 wild-type, null, and mutant (R248W/-) isogenic HCT116 cell lines from Dr. Vogelstein and replaced Figure 3e with the new result obtained using three isogenic cell lines. As the reviewer mentioned, we indicated the amino acid substitution of p53 mutation of the p53 mutant isogenic HCT116 cells in the revised manuscript.

6. Figure 3i – please indicate which stainings were done (most probably similar to panel h) in the figure itself.

We indicated which staining were performed in the revised manuscript.

7. Figure legends 3g and 4h indicate that “immunoblots of extracts from LoVo and HCT116 cell extracts” and “ChIP-qPCR analysis ... in Lovo and HCT116 cells” while in both cases the panels seem to indicate results of only one cell-line.

We corrected the Figure legends in the revised manuscript.

8. Figure 4i+j – it seems that in the ICC images shown for APCKO/p53KO in panel 4i there is still some staining in beta catenin and not in WNT3, while in the quantification shown panel 4j it seems the other way around. Please make sure that there is no confusion in the quantifications, and if indeed the quantifications are correct, please provide more representative pictures which will correspond to the quantifications shown.

We accidently mislabeled the quantification data. The labeling has been corrected in the revised manuscript.

9. Figure 5e – the tumor shown for 5-FU+LGK974 before drug withdrawal looks larger than the tumor for the same treatment group after drug withdrawal, despite the fact that in panel 5d it is shown that the tumors grew to some extent after drug withdrawal. Please show more representative pictures.

The magnifications of the two images were different as shown by the differences in the scale bars. The actual size of the re-grown tumor was larger than the tumor before 5-FU+LGK-974 treatment withdrawal.)

10. Figure 5b-d,6f-i – for the ease of read, please indicate the cell-line/patient derived cell-

line that was used for the experiments also in the figure itself and not only in the figure legend.

As the reviewer commented, we indicated the cell line/patient-derived cell line used for experiments in 5b-d, 6f-q in the Figure and Figure legend of the revised manuscript.

References

1. Yang AD, *et al.* Chronic oxaliplatin resistance induces epithelial-to-mesenchymal transition in colorectal cancer cell lines. *Clin Cancer Res* **12**, 4147-4153 (2006).
2. Voloshanenko O, *et al.* Wnt secretion is required to maintain high levels of Wnt activity in colon cancer cells. *Nat Commun* **4**, 2610 (2013).
3. Phelps RA, *et al.* A two-step model for colon adenoma initiation and progression caused by APC loss. *Cell* **137**, 623-634 (2009).
4. Sanchez-Danes A, *et al.* A slow-cycling LGR5 tumour population mediates basal cell carcinoma relapse after therapy. *Nature* **562**, 434-438 (2018).
5. Shimokawa M, *et al.* Visualization and targeting of LGR5(+) human colon cancer stem cells. *Nature* **545**, 187-192 (2017).
6. Kim NH, *et al.* p53 regulates nuclear GSK-3 levels through miR-34-mediated Axin2 suppression in colorectal cancer cells. *Cell Cycle* **12**, 1578-1587 (2013).
7. Wang J, Shou J, Chen X. Dickkopf-1, an inhibitor of the Wnt signaling pathway, is induced by p53. *Oncogene* **19**, 1843-1848 (2000).
8. Cha YH, Kim NH, Park C, Lee I, Kim HS, Yook JI. MiRNA-34 intrinsically links p53 tumor suppressor and Wnt signaling. *Cell Cycle* **11**, 1273-1281 (2012).
9. Okayama S, *et al.* p53 protein regulates Hsp90 ATPase activity and thereby Wnt signaling by modulating Aha1 expression. *J Biol Chem* **289**, 6513-6525 (2014).
10. Cagatay T, Ozturk M. P53 mutation as a source of aberrant beta-catenin accumulation in cancer cells. *Oncogene* **21**, 7971-7980 (2002).
11. Wang Q, *et al.* The p53 Family Coordinates Wnt and Nodal Inputs in Mesendodermal Differentiation of Embryonic Stem Cells. *Cell Stem Cell* **20**, 70-86 (2017).
12. Comella P, Casaretti R, Sandomenico C, Avallone A, Franco L. Role of oxaliplatin in the treatment of colorectal cancer. *Ther Clin Risk Manag* **5**, 229-238 (2009).
13. Toscano F, *et al.* p53 dependent and independent sensitivity to oxaliplatin of colon cancer cells. *Biochem Pharmacol* **74**, 392-406 (2007).
14. Fujita K, Kubota Y, Ishida H, Sasaki Y. Irinotecan, a key chemotherapeutic drug for metastatic colorectal cancer. *World J Gastroenterol* **21**, 12234-12248 (2015).

15. Takeba Y, *et al.* Irinotecan activates p53 with its active metabolite, resulting in human hepatocellular carcinoma apoptosis. *J Pharmacol Sci* **104**, 232-242 (2007).

REVIEWER COMMENTS

Reviewer #1 (Remarks to the Author):

most of the comments are addressed

Reviewer #2 (Remarks to the Author):

The revision has addressed most of my concerns except one, i.e, would the conclusion of this study be only applied to CRCs with functional APC? The revision shows that LoVo responded very well to Wnt3a stimulation for β -catenin accumulation. It is really at odd against existing literature and our own experience that Wnt3a barely stimulates β -catenin accumulation in cells completely lacking APC. I went back to literature and found out that the original LoVo was heterozygous to an APC mutation. The authors have to either determine LoVo they use is APC-null or limit their conclusion.

Reviewer #3 (Remarks to the Author):

In this revision the authors have attempted to address some of the original comments I made on this manuscript. I had 4 major comments on the first draft and outline them below:

1) The overall conclusions of the study seem counter intuitive. In the introduction the authors outline some known functions of P53 in response to 5FU. All of these are tumour suppressive but this study suggests the opposite, that P53 is an important mediator of disease relapse. Due to the apparent contradiction, more supporting evidence from human patient cohorts is needed. For example, are patients with wild type P53 more likely to relapse? Do they have shorter survival times than P53 mutant patients? What does Wnt signaling look like in these patients?

The authors have not been able to fully address this comments but I concede that due to tumour heterogeneity and variable responses this is challenging. The new data presented supports their hypothesis.

2) The authors demonstrate that following 5FU treatment, organoids exhibit increased stem cell marker expression. What about stem cell function? Clonogenicity assays should be carried out to assess if these are functional stem cells.

I don't feel the authors have not properly addressed this comment. They provide data that Wnt inhibition reduces clonogenicity in Hct116 cells following 5FU treatment. But this does not provide evidence for increased stem cell function upon this treatment. Apcmin organoids should be treated with 5FU and at the timepoint when increase in stem cell marker expression is evident, clonogenicity carried out to test this.

3) What does 5FU do to the organoids and why does Wnt signalling protect Lgr5+ stem cells? The authors should investigate apoptosis and proliferation in the 5FU treated organoids, in particular, are Lgr5+ stem cells more resistant to the effects?

The authors have partially addressed this comment and the dedifferentiation possibility they propose is interesting.

4) Related to this, is Wnt3 sufficient to drive these phenotypes? Does Wnt3 of organoids / cell lines lead to the same Bcat stabilisation and stem cell marker increases? And does this impact on 5FU treatment response?

Point 4 has been addressed.

In summary, the authors have gone some way to addressing my original concerns but I still think there are some additions needed. In particular, addressing my concerns in Point 2.

Reviewer #4 (Remarks to the Author):

The authors successfully answered my queries in a convincing manner. This manuscript indeed presents a very interesting phenomenon and its underlying mechanism, and warrants publication in its current form, in my opinion.

As a side note, regardless of the aforementioned decision, I think that indeed, as reviewer 3 noted, the notion that wild-type (WT) p53 may indirectly contribute to tumor relapse after chemotherapeutic treatment is rather counter-intuitive. As the authors themselves point out, WT p53 is usually thought to be a negative regulator of WNT and a positive predictor of 5-FU response. My interpretation of the current study's findings in light of the aforementioned points, is that while mutant p53 might be a driver of primary drug resistance and WNT activation (and perhaps also CSC activation), WT p53, while predicting initial responsiveness to 5-FU, may cause a relapse and acquired resistance to 5-FU post-treatment. Thus, while both mutant p53 and WT p53 may result in drug resistance and WNT activation, the underlying mechanisms are different. Therefore, I would not exclude the possibility that WNT inhibitors, though probably not LGK-974 but rather cell-intrinsic inhibitors such as beta-catenin inhibitors, might be beneficial to mutant p53 harboring colorectal cancer patients, while LGK-974 may be useful to prevent relapse after 5-FU treatment in WT p53 harboring patients. Although this is not mandatory, and subject to the authors' discretion, explaining these points may prevent confusion and somehow settle the apparent counter-intuitive conclusions from this study.

Point-by-point response to the reviewers

We appreciate the reviewers for their constructive comments for improvement of the manuscript. We addressed the reviewers' concerns regarding the *APC* status of LoVo cells (Reviewer #2) and CSC clonogenicity (Reviewer #3). The revised manuscript is substantially improved by discussing the issues as well as addition of new data in response to the reviewer's comments. We hope that our revised manuscript is now suitable for publication in *Nature Communications*.

Reviewers' comments:

Reviewer #2 (Remarks to the Author):

The revision has addressed most of my concerns except one, i.e, would the conclusion of this study be only applied to CRCs with functional APC? The revision shows that LoVo responded very well to Wnt3a stimulation for β -catenin accumulation. It is really at odd against existing literature and our own experience that Wnt3a barely stimulates β -catenin accumulation in cells completely lacking APC. I went back to literature and found out that the original LoVo was heterozygous to an APC mutation. The authors have to either determine LoVo they use is APC-null or limit their conclusion.

As reviewer indicated, the LoVo cell we used was heterozygous to an *APC* mutation with no loss of heterozygosity¹. We understand the reviewer's concern about this issue; however, we are very cautious with limiting our conclusion to CRCs with functional APC due to the possible involvement of mutant APC and/or APC2 in the WNT3-induced accumulation of β -catenin in *APC*-mutated or-deficient CRCs. Several studies demonstrated that even in the presence of *APC* mutation, downstream signaling remains responsive to WNT ligands^{2,3}. Binding of truncated APCs with β -catenin and key components of β -catenin destruction complex^{2,3} and partial functional redundancy of APC2 with APC as shown by the partial rescue of *APC* deficiency-induced activation of the Wnt/ β -catenin pathway^{4,5} suggest that detailed studies on the involvement of wildtype or mutant APC and APC2 in WNT3-induced β -catenin accumulation may advance our conclusion. However, since our study focuses on the 5-FU-induced activation of cancer stem cells via p53-dependent WNT pathway activation, we believe that further illustration of WNT ligand responsiveness in *APC*-mutated CRCs will improve current understanding of the roles of wildtype and mutant APC and APC2 in CRC. Regarding the importance of the reviewer's comment, we explained these points in the

discussion of the revised manuscript.

Reviewer #3 (Remarks to the Author):

In summary, the authors have gone some way to addressing my original concerns but I still think there are some additions needed. In particular, addressing my concerns in Point 2.

2) The authors demonstrate that following 5FU treatment, organoids exhibit increased stem cell marker expression. What about stem cell function? Clonogenicity assays should be carried out to assess if these are functional stem cells.

I don't feel the authors have not properly addressed this comment. They provide data that Wnt inhibition reduces clonogenicity in Hct116 cells following 5FU treatment. But this does not provide evidence for increased stem cell function upon this treatment. Apcmin organoids should be treated with 5FU and at the timepoint when increase in stem cell marker expression is evident, clonogenicity carried out to test this.

As reviewer commented, to confirm that the increased stem cell marker expression by 5-FU treatment represents the functional characteristics of cancer stem cells, we performed clonogenicity assay using 3D *Apc^{Min/+}/Lgr5^{EGFP}* tumor organoids. To avoid 5-FU effects other than the WNT-dependent CSC activation such as DNA damage, we compared the clonogenicity of *Apc^{Min/+}/Lgr5^{EGFP}* tumor organoid cells after treatment of 5-FU alone or co-treatment with LGK-974 rather than comparing those of control and 5-FU treatment. After 48 hours of 5-FU alone or combinatorial treatment with LGK-974, the time point when cancer stem cell markers are effectively induced by 5-FU and those inductions were inhibited by combinatorial treatment with LGK-974 (Fig. 1f, 5a of revised manuscript), the tumor organoids were dissociated and reseeded with fresh media without drug treatments. Here, initiation of organoid clone formation derived from 5-FU-treated *Apc^{Min/+}* tumor organoids occurred earlier with faster growth rate than that derived from 5-FU and LGK-974 co-treated *Apc^{Min/+}* tumor organoids. (Fig. R1). Taken together, CSC marker expression regulation by 5-FU or 5-FU and LGK-974 co-treatment in this study is associated with functional characteristics of colorectal CSCs. We addressed this issue in the revised manuscript (Supplementary Fig. 5a-c).

Fig. R1 LGK-974 suppressed the clonogenicity of 5-FU-treated tumor organoids derived from *Apc^{Min/+}* mice. Tumor organoids derived from *Apc^{Min/+}* mice were treated with 5-FU (1.5 $\mu\text{g/ml}$, 48 h) alone or co-treated with LGK-974 (5 μM , 48 h). After 48 h, tumor organoids were dissociated and passaged as single cells in fresh media without drug treatment, and the formation and growth of organoids were observed for 12 days. **a** scheme of the assay. **b** Representative bright-field images at each time point by EVOS microscope (Invitrogen). Scale bar=650 μm . **c** Growth of tumor organoids derived from 5-FU alone or co-treated with LGK-974 *Apc^{Min/+}* tumor organoids were measured using Cell Titer-Glo[®] Luminescent Cell Viability Assay at indicated days. Data are mean \pm s.d., two-sided Student's t-test, ** $p < 0.01$, *** $p < 0.001$.

Reviewer #4 (Remarks to the Author):

The authors successfully answered my queries in a convincing manner. This manuscript indeed presents a very interesting phenomenon and its underlying mechanism, and warrants

publication in its current form, in my opinion.

As a side note, regardless of the aforementioned decision, I think that indeed, as reviewer 3 noted, the notion that wild-type (WT) p53 may indirectly contribute to tumor relapse after chemotherapeutic treatment is rather counter-intuitive. As the authors themselves point out, WT p53 is usually thought to be a negative regulator of WNT and a positive predictor of 5-FU response. My interpretation of the current study's findings in light of the aforementioned points, is that while mutant p53 might be a driver of primary drug resistance and WNT activation (and perhaps also CSC activation), WT p53, while predicting initial responsiveness to 5-FU, may cause a relapse and acquired resistance to 5-FU post-treatment. Thus, while both mutant p53 and WT p53 may result in drug resistance and WNT activation, the underlying mechanisms are different. Therefore, I would not exclude the possibility that WNT inhibitors, though probably not LGK-974 but rather cell-intrinsic inhibitors such as beta-catenin inhibitors, might be beneficial to mutant p53 harboring colorectal cancer patients, while LGK-974 may be useful to prevent relapse after 5-FU treatment in WT p53 harboring patients. Although this is not mandatory, and subject to the authors' discretion, explaining these points may prevent confusion and somehow settle the apparent counter-intuitive conclusions from this study.

We appreciate the reviewer's suggestion and agree with the comment that explaining these points may prevent confusion. We addressed these points in the discussion of the revised manuscript.

References

1. Rowan AJ, *et al.* APC mutations in sporadic colorectal tumors: A mutational "hotspot" and interdependence of the "two hits". *Proc Natl Acad Sci U S A* **97**, 3352-3357 (2000).
2. Voloshanenko O, *et al.* Wnt secretion is required to maintain high levels of Wnt activity in colon cancer cells. *Nat Commun* **4**, 2610 (2013).
3. Hochman G, Halevi-Tobias K, Kogan Y, Agur Z. Extracellular inhibitors can attenuate tumorigenic Wnt pathway activity in adenomatous polyposis coli mutants: Predictions of a validated mathematical model. *PLoS One* **12**, e0179888 (2017).

4. van Es JH, *et al.* Identification of APC2, a homologue of the adenomatous polyposis coli tumour suppressor. *Curr Biol* **9**, 105-108 (1999).
5. Saito-Diaz K, *et al.* APC Inhibits Ligand-Independent Wnt Signaling by the Clathrin Endocytic Pathway. *Dev Cell* **44**, 566-581 e568 (2018).

REVIEWERS' COMMENTS:

Reviewer #2 (Remarks to the Author):

I am satisfied with the addition of the discussion to limit the scope of its conclusion.

Reviewer #3 (Remarks to the Author):

The authors have addressed my comments and I would now recommend publication.

Final Reply to Reviewers:

Reviewers' Comments:

Reviewer #2 (Remarks to the Author):

I am satisfied with the addition of the discussion to limit the scope of its conclusion.

Reviewer #3 (Remarks to the Author):

The authors have addressed my comments and I would now recommend publication.

Point by point response to the reviewers:

We appreciate that the reviewer was positive on our updated data and gave favorable comments.